# Interpretable Probability Estimation with LLMs via Shapley Reconstruction

## Abstract

Large Language Models (LLMs) demonstrate potential to estimate the probability of uncertain events, by leveraging their extensive knowledge and reasoning capabilities. This ability can be applied to support intelligent decision-making across diverse fields, such as financial forecasting and preventive healthcare. However, directly prompting LLMs for probability estimation faces significant challenges: their outputs are often noisy, and the underlying predicting process is opaque. In this paper, we propose **PRISM: Probability Reconstruction via Shapley Measures**, a framework that brings transparency and precision to LLM-based probability estimation. PRISM decomposes an LLM's prediction by quantifying the marginal contribution of each input factor using Shapley values. These factor-level contributions are then aggregated to reconstruct a calibrated final estimate. In our experiments, we demonstrate PRISM improves predictive accuracy over direct prompting and other baselines, across multiple domains including finance, healthcare, and agriculture. Beyond performance, PRISM provides a transparent prediction pipeline: our case studies visualize how individual factors shape the final estimate, helping build trust in LLM-based decision support systems.

## 1 Introduction

Estimating the probability of uncertain events (Berger, 2013; Winkler et al., 2019) is a critical task for intelligent decision makings in various domains such as financial investment (Lathief et al., 2024), healthcare (Rajkomar et al., 2019), and emergency management (Rostami-Tabar and Hyndman, 2025). However, in many real-world scenarios, either high-quality datasets are unavailable or mature Machine Learning (ML) techniques are lacking. Besides, there could also appear temporarily unexpected factors which are not considered when people building datasets (Chowdhury et al., 2021). As an example, in business analytics (Raghupathi and Raghupathi, 2021), people may encounter a variety of diverse estimation tasks, such as determining whether the price, production or market demand of a certain product will increase or not (Fildes et al., 2022). It could be difficult for them to collect sufficient task-specific data and build reliable predictive models promptly. In this scenario, the applicability of traditional ML approaches are severely limited.

As a potential solution, Large Language Models (LLMs) offer a promising alternative to address this challenge (Feng et al., 2024; Sui et al., 2024; Chung et al., 2024). They incorporate extensive world knowledge and exhibit strong reasoning capabilities (Wei et al., 2022), making them particularly valuable in data or model scarce settings. However, directly prompting LLMs to probability estimation still faces tremendous challenges: (i) LLMs typically produce noisy probability estimates that lack accuracy and stability. As an instance, according to the study (Nafar et al., 2025), when LLMs are asked to predict the same event in different forms, i.e., "whether it will happen" and "whether it will not happen", LLMs could provide conflicting answers. (ii) The "black-box" generative process provides little transparency regarding how each individual factor contributes to the final prediction, making the results difficult to interpret. As illustrated in Figure 1, when asked to predict the likelihood of a person having a certain disease, LLMs will not explicitly show how much weight each factor contributes to final prediction, but outputs a single score (with a partial explanation). It makes the final outcome difficult to interpret and less trustworthy.

To overcome the challenges, we propose a novel framework **Probability Reconstruction via Shapley Measures (PRISM)**, inspired by Shapley Value for ML explanation (Lundberg and Lee, 2017). Specifically, we decompose the estimation task into quantifying the marginal contribution of each

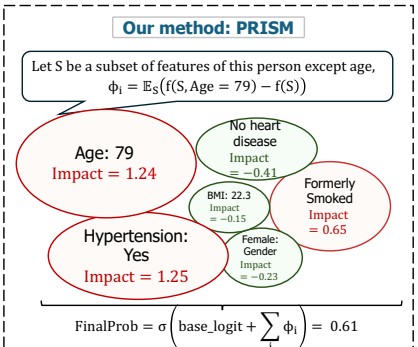 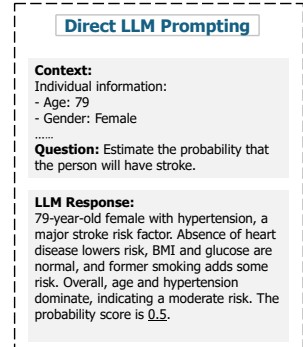

Figure 1: Illustration of direct LLM prompting and PRISM. PRISM first estimates Shapley values (factor contributions) and aggregates them to reconstruct final probability. We use red to represent the factors found by PRISM to have positive contribution to the positive outcome (have a stroke), green are factors found to be negative. The size reflects the contribution's absolute value. $f(\cdot)$ is from LLM prediction and S is a background set, $\sigma(\cdot)$ is the sigmoid function (see details in Section 3).

factor independently. As illustrated in Figure 1, consider the task of predicting whether a person will experience a stroke. For a given factor such as "Age=79", we compare the LLM's estimated probability when the model has access to this factor versus when it does not have access to it, paired with a random subset S of the remaining factors (see Section 3 for details). The *marginal contribution of a factor*, referred to as its *Shapley value*, is computed by averaging over different subsets S. These values capture both the direction and the intensity of the features' contribution to the prediction outcome. However, unlike traditional Shapley methods which solely focus on model output attribution (Lundberg and Lee, 2017), PRISM goes further: we reconstruct a new probability estimate by aggregating the factor contributions. For example, in Figure 1, the aggregated contributions yield a probability of 0.61, and we can clearly interpret this estimate originates from key risk-increasing factors such as advanced age and having hypertension. This process can make the prediction process transparent and allow human users to diagnose and interpret the model's reasoning.

In our experiments, we validate the effectiveness of our proposed method under various settings. In Section 4.1, we compare PRISM with other LLM-based probability estimation methods, on binary classification tasks under benchmark tabular datasets (such as Adult Census Income (Becker and Kohavi, 1996), Heart Disease Prediction (Chicco and Jurman, 2020), Stroke Prediction (Kaggle, 2021) and Loan Default Prediction (LendingClub, 2007–2018)). We find that PRISM demonstrates higher prediction performance than directly LLM prompting and other representative baselines. Furthermore, in Section 4.2, we evaluate PRISM on real-world prediction tasks involving textual and numerical inputs. For instance, we predict whether the price of an agricultural commodity will rise in the next year using information from the previous year's annual report, and we forecast the outcomes of English football matches based on pre-match reports. Together, these studies demonstrate PRISM's ability to handle heterogeneous, context-rich factors—well beyond simple tabular data. Overall, the *experiments highlight PRISM's potential as a practical and reliable tool* for a wide range of real-world prediction problems.

## 2 RELATED WORKS

### 2.1 PROBABILITY ESTIMATION VIA TRADITIONAL ML AND LLMS

Traditional machine learning models such as Bayesian inference (Berger, 2013), multilayer perceptrons (MLPs) (Rumelhart et al., 1986) and decision trees (Quinlan, 1986) have long been widely used for probability estimation, typically cast as binary classification tasks on tabular data. These methods often rely on large amount of data and well-designed models. To overcome this limitation, recent works have explored using Large Language Models (LLMs) for probability estimation by leveraging their rich prior knowledge. In this paper, we focus on LLM-based probability estimation **under zero-shot setting** which makes estimations only based on the LLM's own knowledge. Under this setting, previous studies validate the potential especially for tabular format datasets (Sui et al., 2024; Chung et al., 2024; Ren et al., 2025; Xie et al., 2024). However, although many of these methods show decent estimation precision, their estimation process usually relies on direct LLM

prompting. Thus, it can be difficult for the users to exactly measure the contribution of each factor to the final prediction, different from the case in traditional ML models. As a more relevant paper to our work, BIRD (Feng et al., 2024) is proposed to integrate Bayesian networks (Berger, 2013) to explicitly calculate the probability of the prediction outcome conditioning on each factor. However, it assumes the factors are independent to each other, and it has to transform the factors into categorical values. Notably, beyond the zero-shot setting, there are related studies focusing on few-shot prediction with LLMs (Hegselmann et al., 2023; Brown et al., 2020). However, in such cases, the interpretation can become more complicated, as it should disentangle whether a prediction arises from the own knowledge of LLMs or from the provided demonstrations. Therefore, we leave the exploration of few-shot prediction tasks to future work.

## 2.2 EXPLAINABLE PROBABILITY ESTIMATION

For traditional ML explanation, Shapley-value (Shapley et al., 1953; Lundberg and Lee, 2017; Štrumbelj and Kononenko, 2014) is widely used to attribute predicted probabilities to input factors. In general, it quantifies each factor's marginal contribution to the final prediction outcome. Building on this line, recent works adapt Shapley values to explain the behavior of LLMs. For instance, TokenSHAP (Goldshmidt and Horovicz, 2024) estimates token-level contributions to LLM outputs in general question answering tasks. SyntaxSHAP (Amara et al., 2024) moves beyond token-level attributions by capturing the contribution of higher-level syntactic units. Unlike these studies, our work specifically targets on probability estimation tasks, where we quantify the contributions of each "influencing factor" in the probability estimation. Besides,, our approach goes beyond conventional LLM or ML interpretation tools. We reconstruct new probability estimates through a principled approach of factor aggregation, rather than only interpreting the original model outputs.

## 3 METHODOLOGY

In this section, we formally introduce and describe our proposed method Probability Reconstruction via Shapley Measures (PRISM). In Section 3.1, we first introduce the necessary notations and definitions, with particular emphasis on the Shapley value and its importance for ML model explanations. Section 3.2 then provides a full description of the PRISM algorithm, clarifying how factor contributions are combined to reconstruct probabilities. Finally, Section 3.3 introduces an efficient variant of PRISM specifically designed for tabular tasks, enabling faster computation in such scenarios.

### 3.1 DEFINITIONS AND NOTATIONS

**Notations.** In our study, for a probability estimation task, we denote an **instance** by $x = (x_1, \ldots, x_m)$ and its true outcome by $y_{true} \in \{0, 1\}$, where each $x_i$ is a factor that can influence the outcome. We let $f(\cdot)$ denote a general model output, which may come from either a traditional ML binary classifier or from LLM-based predictions. Depending on the setting, $f(\cdot)$ can represent either the predicted probability or the model logit (before sigmoid function). In general, it is desirable to obtain the estimation $f(x)$ to be close or aligned to $y_{true}$.

In our paper, an important concept is Shapley value (Shapley et al., 1953) that applied in ML prediction explanation (Lundberg and Lee, 2017). In the next, we first introduce the basic definition of Shapley value (for ML models) in general, and we discuss one important property of it.

**Definition 1** (Shapley value). *Let* $\mathcal{I} = \{1, 2, \ldots, m\}$ *denote the index set of factors, and let* $x = (x_1, \ldots, x_m)$. *The Shapley value of factor* $i \in \mathcal{I}$ *for instance* $x$ *with respect to model* $f$ *is:*

$$\phi_i(f, x) = \sum_{S \subseteq \mathcal{I} \setminus \{i\}} \frac{|S|! \, (m - |S| - 1)!}{m!} \left[ f(x_{S \cup \{i\}}) - f(x_S) \right], \tag{1}$$

*where each* $S$ *is a subset of factors* $\mathcal{I}$ *excluding* $i$*, and the sum is taken over all* $S \subseteq \mathcal{I} \setminus \{i\}$*,* $f(x_S)$ *is the model output when only the factors in* $S$ *are used (and* $f(x_{S \cup \{i\}})$ *is defined analogously). In our paper, we name each* $S$ *as a "**background set**".*

Intuitively, Shapley value $\phi_i(f, x)$ measures the *marginal contribution* of factor $i$ to the model output, by comparing the model's output when factor $i$ is "included in the background set $S$" versus when it is "not included in the background set $S$", across all possible cases of $S$. In this way, one can interpret the model's prediction by separately examining the contribution of each individual factor. In practice, for traditional ML, the difference $\left( f(x_{S \cup \{i\}}) - f(x_S) \right)$ can be computed either by retraining models on different subsets of factors (Lipovetsky and Conklin, 2001), or by approximating the

effect of missing factors using training data (Štrumbelj and Kononenko, 2014). Next, we discuss one important property of Shapley value in traditional MLs which inspires our method.

**Property 1** (Additivity (Lundberg and Lee, 2017)). *Let $\phi_0 = \mathrm{f}(\mathrm{x}_\emptyset)$ be the expected model output when no factor information is available. Then, for models with deterministic scalar outputs, the sum of $\phi_0$ and the Shapley value of the factors $\phi_\mathrm{i}(\mathrm{f}, \mathrm{x})$ must **reconstruct** the model prediction itself.*

$$\mathrm{f}(\mathrm{x}) = \phi_0 + \sum_{\mathrm{i}=1}^{\mathrm{m}} \phi_\mathrm{i}(\mathrm{f}, \mathrm{x}). \tag{2}$$

*That is, the explanation exactly matches the original model output.*

It suggests for traditional ML models, the output $\mathrm{f}(\mathrm{x})$ can be decomposed to the sum of Shapley values and the base logit $\phi_0$. However, Property 1 does not extend easily to Large Language Models (LLMs). This is because, when LLMs make predictions for uncertain events, they express their probability estimates through generated text (see Figure 1 (right)). Such expressed probabilities differ from the direct model outputs, such as the token probabilities produced by the LLM architectures.

### 3.2 PRISM: PROBABILITY RECONSTRUCTION VIA SHAPLEY MEASURES

Nevertheless, inspired by Property 1, we devise our new estimation framework: we first obtain the Shapley values for each factor and then aggregate them to reconstruct a new prediction outcome. In the next, we elaborate two important steps to obtain the Shapley values through LLMs.

**Comparative LLM Prompting.** Given a factor of interest, such as "i = Age is 79" in the stroke prediction task, we first sample multiple background sets S from $\mathcal{I} \setminus \{\mathrm{i}\}$ (following the permutation sampling strategy in the next part). Each S contains information of several factors that different from i, such as "hypertension, female and BMI=22.3" illustrated in Figure 2. Next, for a given background set S, we prompt the LLM to evaluate both $\mathrm{x}_{\mathrm{S} \cup \{\mathrm{i}\}}$ and $\mathrm{x}_\mathrm{S}$ in the same query.

> **Case A:**
> S = {has hypertension, female, BMI 22.3}
>
> **Case B:**
> S ∪ {i} = {has hypertension, female, BMI 22.3, Age = 79}
>
> **Question:** Estimate the probability of each person that having stroke?

Figure 2: Comparative Prompting.

We denote $\mathrm{p}(\cdot)$ as the LLM estimated probability for a given instance (e.g., by responding to a question as in Figure 2). Then, if we let $\mathrm{f}(\cdot) = \sigma^{-1}(\mathrm{p}(\cdot))$, where $\sigma^{-1}(\cdot)$ is the inverse sigmoid function, we have:

$$\mathrm{f}(\mathrm{x}_{\mathrm{S} \cup \{\mathrm{i}\}}) - \mathrm{f}(\mathrm{x}_\mathrm{S}) = \sigma^{-1}\big(\mathrm{p}(\mathrm{x}_{\mathrm{S} \cup \{\mathrm{i}\}})\big) - \sigma^{-1}\big(\mathrm{p}(\mathrm{x}_\mathrm{S})\big), \tag{3}$$

This calculates the difference term in Shapley value defined in Eq.(1). We argue that: the single estimates derived by directly LLM prompting may not be exact, but the relative relation between the two cases can be well-captured. Recent studies also find that LLMs tend to perform more reliably on ranking or comparison tasks than absolute probability estimation (Qin et al., 2023; Liu et al., 2024), which supports the validity of our design.

**Permutation Sampling.** In PRISM, at each iteration we sample a background set S using the permutation sampling rule (Shapley et al., 1953; Štrumbelj and Kononenko, 2014). Concretely, we generate a random permutation of all factors in $\mathcal{I}$ including i, and choose S as the set of factors that precede i in this order. Then, we directly average the pair-wise differences (from Eq.(3)) across multiple (e.g., K times) such samplings to approximate the Shapley value:

$$\phi_\mathrm{i} = \frac{1}{\mathrm{K}} \sum_{\mathrm{k}=1}^{\mathrm{K}} \left[ \sigma^{-1}\big(\mathrm{p}(\mathrm{x}_{\mathrm{S}^{(\mathrm{k})} \cup \{\mathrm{i}\}})\big) - \sigma^{-1}\big(\mathrm{p}(\mathrm{x}_{\mathrm{S}^{(\mathrm{k})}})\big) \right], \tag{4}$$

The equivalence of Eq.(4) and Eq.(1) is shown in (Shapley et al., 1953). In practice, it is necessary to sample S for multiple times, ensuring each background factor is considered with a non-negligible probability. This will lead to a more comprehensive estimation by accounting for possible factor interactions (see Section 4.3 for further discussion).

Finally, we set $\phi_0$ as the base logit, either from the population average (e.g., $\phi_0 = \sigma^{-1}$(Average stroke prevalence), or simply set to 0 when no prior knowledge is available in other tasks. We then reconstruct the probability estimation as:

$$\mathrm{p}_{\text{PRISM}}(\mathrm{x}) = \sigma(\phi_0 + \sum_{\mathrm{i}=1}^{\mathrm{m}} \phi_\mathrm{i}). \tag{5}$$

We provide the detailed algorithm sketch in Appendix A.1.

### 3.3 TABULAR-PRISM

In this subsection, we discuss a variant of PRISM, when the factors can be represented as single values or categories and be integrated in tabular datasets. For such tasks, instead of calculating the difference term in Eq.(3) for each S in a separate LLM query, we can instead consider multiple background sets in **one single query**. Refer to Figure 3a, we can present $x_S$ and $x_{S \cup \{i\}}$ for different S in the same table (the adjacent rows are from the same S). Then, we prompt LLMs to evaluate each of them in one query. In this way, the query and token efficiency can be significantly improved. However, a challenge is that: if we leave those factors that are not presented as blank, "unknown" or "not provided", it will introduce bias to the LLM judgment. For example, LLM can over-estimate a person's risk of certain disease only because some information is "unknown".

To solve this issue, we leverage the strategy to introduce a **reference instance**, which is also used in Shapley ML explanation (Štrumbelj and Kononenko, 2014). Specifically, we let r be a reference instance with each factor to be a fixed value (typically obtained from population average or majority). Then, we use the information of this reference instance to impute missing factors, which is illustrated in Figure 3b, and we calculate the Shapley values (with a new definition) based on this new table.

| Age | Hyper. | Heart Disease | BMI |
|---|---|---|---|
|  | Yes |  | 22.3 |
| 79 | Yes |  | 22.3 |
|  |  | No | 22.3 |
| 79 |  | No | 22.3 |
|  | Yes |  |  |
| 79 | Yes |  |  |

| Age | Hyper. | Heart Disease | BMI |
|---|---|---|---|
| 40 | Yes | No | 22.3 |
| 79 | Yes | No | 22.3 |
| 40 | No | No | 22.3 |
| 79 | No | No | 22.3 |
| 40 | Yes | No | 24.0 |
| 79 | Yes | No | 24.0 |

(a) With blanks.   (b) With reference.

Figure 3: When calculating Shapley value of the factor "Age=79", we put multiple S in one table. Factor values from reference instances are noted in green.

**Definition 2.** *For background set S, we define the model output of the imputed sample as* $v_r(S) = f([x_S, r_{\bar{S}}])$, *where* $\bar{S} = \mathcal{I} \setminus S$, *which means the factors in S are provided by* x, *and the remaining are provided by* r. *Then, for each factor* $x_i$, *the (reference-specific) Shapley value is*

$$\phi_i^{(r)} = \sum_{S \subseteq \mathcal{I} \setminus \{i\}} \frac{|S|! \, (m - |S| - 1)!}{m!} \left( v_r(S \cup \{i\}) - v_r(S) \right). \tag{6}$$

Essentially, this new definition has a different interpretation compared to the original definition in Eq.(1). It emphasizes the comparison to the reference sample. For example, a large positive $\phi_i^{(r)}$ for "Age = 79" suggests: compared with "Age = 40" (from reference sample), the factor "Age = 79" increases the risk of having stroke greatly. Despite different interpretations, the following proposition can still allow us to reconstruct predictions from these Shapley values.

**Proposition 1.** *Fix an instance* x *and a single reference sample* r. *Let* $\phi_i^{(r)}$ *be the Shapley value in Definition 2. Then, for models with deterministic scalar outputs, with* $\phi_0^{(r)} = v_r(\emptyset)$, *we still have:*

$$f(x) = v_r(\mathcal{I}) = \phi_0^{(r)} + \sum_{i=1}^{m} \phi_i^{(r)} \tag{7}$$

Similar to Property 1, this proposition suggests that: in LLMs, we can also reconstruct a new estimation by aggregating Shapley values, following the rule: $p_{PRISM}(x) = \sigma(\phi_0^{(r)} + \sum_{i=1}^{m} \phi_i^{(r)})$. We defer the detailed algorithm sketch and proof of Proposition 1 in Appendix A.1 and Appendix A.2.

## 4 EXPERIMENT AND DISCUSSION

In this section, we conduct experiments to validate the effectiveness of PRISM. In Section 4.1, we compare the estimation performance of PRISM and baselines on benchmark (tabular) datasets. Meanwhile, we provide case analysis to visualize the interpretation outcome of PRISM. In Section 4.2, we demonstrate that PRISM can also be applied in real-world estimation tasks which cannot be easily formed into tabular format. Finally, Section 4.3 provides ablation studies about whether PRISM can handle feature interactions (Hall, 1999), and discuss its computational efficiency.

### 4.1 PROBABILITY ESTIMATION ON BENCHMARK TABULAR DATASETS

We first evaluate our method focused on benchmark **tabular** datasets. This is because probability estimation tasks are more frequently built in tabular format, for traditional ML studies. Therefore,

we have various datasets with sufficient samples for fair and comprehensive comparisons. We later discuss scenarios beyond tabular format in Section 4.2.

**Experiment setup.** We involve four representative tabular datasets ranging from disease prediction, income prediction, and loan credit estimation. In details, Adult Census Income (Becker and Kohavi, 1996) is to predict whether a person has annual income over $50K, based on their occupation, education and family information. Stroke (Kaggle, 2021) and Heart Disease (Chicco and Jurman, 2020) predict whether a certain disease will appear based on the patients' health status. Lending (Lending-Club, 2007–2018) predicts whether a loan default will happen, based on the loan application records and applicants' personal information. More details and pre-process procedures are in Appendix B.1.

In our study, we majorly consider the zero-shot settings, where the estimations are solely relied on the LLM's own knowledge. For our method, we implement Tabular-PRISM (in Section 3.3) and we sample the background set S for 10 times (for each Shapley value) [1]. We also consider the baselines:

- Directly prompting LLMs to predict likelihood **levels**. For example, we ask LLM to choose one in the options from "very unlikely" to "very likely". We also try multiple shots to obtain the self-consistency result (Wang et al., 2022) by making votes. They are denoted as "1shot_level, 5shot_level, 10shot_level" in Table 1.
- Directly prompting LLMs to predict likelihood **scores**, which are probability scores between [0-1], and we obtain the self-consistency result by taking their average. They are denoted as "1shot_score, 5shot_score, 10shot_score" in Table 1
- *Contrast* (motivated by Nafar et al. (2025)) asks about the likelihood in the positive and the negated question form, and then unify the answers to get the final estimation.
- *BIRD* (Feng et al., 2024) builds Bayesian networks to evaluate the probability of the estimation outcome conditioning on various (categorized) factors.
- We also add In-Context Learning (ICL, Brown et al. (2020)), which is beyond zero-shot setting. We randomly select 5 positive and 5 negative samples as demonstrations for each instance (or 10 positive and 10 negative respectively).

For each setting and method, we conduct experiments on GPT-4.1-mini (OpenAI, 2024) and Gemini-2.5-Pro (Google DeepMind, 2025). All runs use temperature 1.0 under default settings.

**Experiment result.** In our result shown in Table 1, for each dataset, we randomly choose 300 test samples (150 positive and 150 negative), and we report AUROC (shown as "ROC" in Table 1), AUPRC (shown as "PRC"), and the best F1 (when select the threshold for maximized TPR + TNR). From the result, we can see the PRISM consistently demonstrates reliable estimation performance. Specifically, PRISM presents highest performance or it is close to the strongest baselines in most datasets, including Adult Census, Heart Disease and Lending, under both LLMs. In these datasets, we see PRISM has more obvious improvement in AUROC and AUPRC than F1-scores. This suggests that the baselines can sometimes effectively separate positive and negative samples when an appropriate threshold is chosen, whereas the high AUROC and AUPRC of PRISM indicate its strong discriminative ability across all possible thresholds. Under Stroke Dataset, our method is comparable to strong baselines or slightly lower than those strong baselines. For example, under Gemini-2.5-pro, "1shot_score" has higher AUROC, AUPRC and F1 than PRISM. We conjecture it may be because the LLM itself has well-educated knowledge in the relevant domain. However, PRISM demonstrates stable performance across various datasets and shows competency against the strongest baselines, if not surpassing them. In Appendix C, we show PRISM's predicted probability also has a good calibration (Bella et al., 2010) if a proper base logit is selected.

For the baseline methods, we find "score" based LLM prompting are generally better than "level" based prompting. The most comparable baseline with PRISM is "5shot_score" and "10shot_score". However, we would like to argue that: the multiple-query strategy (or namely Self-Consistency) is less interpretable than single-query in practice, as they rely on aggregating multiple diverse estimation paths, making it difficult for a uniform assessment of the factor contribution. Besides, BIRD

---

[1]We set the factor values in the reference instance r to be around the population average (for continuous values), and population majority (for categorical values). We query an LLM for the base logit $\phi_0^{(r)}$. In practice, one can choose $\phi_0^{(r)}$ from more reliable sources such as historical statistics or human experts, and the selection of $\phi_0^{(r)}$ will not impact the later evaluation metrics, including AUROC, AUPRC and F1.

| Model / Method | Adult Census | | | Heart Disease | | | Stroke | | | Lending | | |
|---|---|---|---|---|---|---|---|---|---|---|---|---|
| | ROC | PRC | F1 | ROC | PRC | F1 | ROC | PRC | F1 | ROC | PRC | F1 |
| **GPT-4.1-mini** | | | | | | | | | | | | |
| 1shot_level | 0.777 | 0.773 | 0.709 | 0.722 | 0.706 | 0.615 | 0.767 | 0.694 | 0.726 | 0.629 | 0.606 | 0.478 |
| 5shot_level | 0.795 | 0.779 | 0.701 | 0.759 | 0.723 | 0.664 | 0.780 | 0.716 | 0.730 | 0.617 | 0.576 | 0.648 |
| 10shot_level | 0.799 | 0.779 | 0.701 | 0.759 | 0.718 | 0.672 | 0.792 | 0.727 | 0.732 | 0.627 | 0.591 | 0.443 |
| 1shot_score | 0.795 | 0.792 | 0.709 | 0.799 | 0.761 | 0.772 | 0.804 | 0.763 | 0.753 | 0.612 | 0.600 | 0.558 |
| 5shot_score | 0.819 | 0.820 | 0.755 | 0.807 | 0.779 | 0.782 | **0.816** | 0.783 | 0.780 | 0.629 | 0.621 | 0.612 |
| 10shot_score | 0.816 | 0.820 | 0.734 | 0.806 | 0.776 | **0.793** | 0.813 | 0.781 | 0.785 | 0.636 | **0.631** | 0.621 |
| ICL-5+5 | 0.807 | 0.788 | 0.707 | 0.803 | 0.761 | 0.762 | 0.801 | 0.751 | 0.775 | 0.598 | 0.567 | 0.668 |
| ICL-10+10 | 0.754 | 0.758 | 0.645 | 0.776 | 0.744 | 0.760 | 0.788 | 0.736 | 0.763 | 0.591 | 0.569 | 0.642 |
| Contrast | 0.790 | 0.813 | 0.697 | 0.769 | 0.729 | 0.738 | 0.790 | **0.813** | 0.697 | 0.485 | 0.543 | 0.169 |
| BIRD | 0.813 | 0.804 | 0.730 | 0.777 | 0.748 | 0.712 | 0.778 | 0.740 | 0.729 | 0.610 | 0.564 | 0.533 |
| PRISM (Ours) | **0.851** | **0.874** | **0.770** | **0.816** | **0.799** | **0.793** | 0.814 | 0.783 | **0.790** | **0.655** | 0.626 | **0.671** |
| **Gemini-2.5-Pro** | | | | | | | | | | | | |
| 1shot_level | 0.818 | 0.797 | 0.699 | 0.732 | 0.668 | 0.780 | 0.793 | 0.744 | 0.720 | 0.555 | 0.538 | 0.518 |
| 5shot_level | 0.822 | 0.794 | 0.694 | 0.758 | 0.686 | 0.794 | 0.804 | 0.749 | 0.727 | 0.541 | 0.526 | 0.511 |
| 10shot_level | 0.821 | 0.797 | 0.688 | 0.738 | 0.663 | 0.791 | 0.803 | 0.745 | 0.716 | 0.564 | 0.543 | 0.571 |
| 1shot_score | 0.855 | 0.876 | 0.767 | 0.812 | 0.739 | 0.797 | **0.836** | 0.812 | 0.802 | 0.536 | 0.538 | 0.465 |
| 5shot_score | 0.864 | 0.879 | 0.823 | 0.816 | 0.749 | 0.795 | 0.834 | 0.810 | **0.804** | 0.529 | 0.534 | 0.533 |
| 10shot_score | 0.864 | 0.878 | **0.826** | 0.815 | 0.753 | **0.799** | 0.835 | **0.814** | 0.803 | 0.528 | 0.526 | 0.547 |
| ICL-5+5 | 0.842 | 0.834 | 0.744 | 0.834 | 0.785 | 0.770 | 0.815 | 0.767 | 0.728 | 0.615 | 0.576 | 0.575 |
| ICL-10+10 | 0.812 | 0.793 | 0.722 | 0.807 | 0.756 | 0.739 | 0.818 | 0.776 | 0.738 | 0.626 | 0.584 | **0.690** |
| Contrast | 0.835 | 0.830 | 0.757 | 0.831 | 0.779 | 0.789 | 0.806 | 0.768 | 0.727 | 0.593 | 0.588 | 0.520 |
| BIRD | 0.799 | 0.794 | 0.754 | 0.810 | 0.767 | 0.762 | 0.803 | 0.779 | 0.735 | 0.555 | 0.528 | 0.625 |
| PRISM (Ours) | **0.876** | **0.893** | 0.826 | **0.844** | **0.832** | 0.787 | 0.826 | 0.788 | 0.783 | **0.654** | **0.614** | 0.685 |

Table 1: Performance comparison across various tabular datasets. Best results are in dark red.

| Factor | Case 1 | | Case 2 | | Case 3 | | Case 4 | |
|---|---|---|---|---|---|---|---|---|
| | Value | Shapley | Value | Shapley | Value | Shapley | Value | Shapley |
| Gender | Female | -0.08 | Female | 0.00 | Male | 0.00 | Female | 0.00 |
| Age | 82 | 1.17 | 52 | 0.57 | 68 | 1.15 | 29 | -0.77 |
| Hypertension | Yes | 1.64 | Yes | 1.41 | No | 0.00 | No | 0.00 |
| Heart Disease | Yes | 0.66 | No | 0.00 | Yes | 1.17 | No | 0.00 |
| Marital Status | Never Married | 0.00 | Ever Married | 0.00 | Ever Married | 0.00 | Ever Married | 0.00 |
| Work Type | Government job | 0.00 | Private sector | 0.00 | Private sector | 0.00 | Private sector | 0.00 |
| Residence Type | Rural | 0.00 | Urban | 0.00 | Urban | 0.00 | Urban | 0.00 |
| Glucose Level | 84.03 | 0.00 | 94.98 | 0.51 | 223.83 | 0.62 | 116.98 | 0.98 |
| BMI | 25.60 | 0.00 | 23.80 | 0.00 | 31.90 | 0.53 | 23.40 | 0.00 |
| Smoking Status | Smokes | 0.85 | Never smoked | 0.00 | Formerly smoked | 0.60 | Never smoked | 0.00 |
| *Sum Shapley* | 4.240 | | 2.490 | | 4.070 | | 0.210 | |
| *Sum logit* | -0.355 | | -2.105 | | -0.525 | | -4.385 | |
| *Predicted prob* | 0.413 | | 0.109 | | 0.372 | | 0.012 | |
| *True label* | Yes | | No | | Yes | | No | |

Table 2: Shapley values for four instances in Stroke dataset. Reference instance: *gender=Male; age=40; hypertension=No; heart disease=No; marital status=Never Married; residence=Rural; average glucose=90.0; BMI=24.0; work type=Private; smoking status=never smoked*. A single base logit is shared across cases, $\phi_0 = \sigma^{-1}(0.01)$=-4.5951.

is the only method with explicit interpretable structure among the baselines. Compared to BIRD, PRISM has obviously higher performance across different settings.

**Examination of Interpretations.** In Table 2, we present instances under Stroke dataset to help understand the interpretation process of PRISM. In detail, we present the factor values as well as the estimated Shapley values (Eq.(6)). Notably, the Shapley value here represents the relative contribution of each factor, comparing to the reference instance (see Section 3.3). For example, in Case 1, a Shapley value of 0.66 for "Heart Disease" suggests: compared with "No Heart Disease" (from reference instance), this factor increases the risk of having stroke. Similarly, in Case 2, because the person does not have heart disease which is the same as reference instance, the Shapley value is 0. In Appendix C, we provide the interpretation results similar to Table 2 of other datasets.

## 4.2 PROBABILITY ESTIMATION ON UNSTRUCTURED DATA

Beyond tabular data, there are often cases where the influencing factors cannot be easily transformed into single values to be inputted into tables. For example, the factors can be in form of descriptive facts that are extracted from unstructured sources such as news articles, financial reports, or social media. Probability estimation in this scenario is also of great importance. We conduct two case studies to demonstrate the applicability of PRISM (the version in Section 3.2) in such setting.

**Predicting apple price**. Our first task is to determine "whether the price of apple will increase in 2025 compared to 2024?", based on (U.S. Apple Association annual report 2024). Such a task may later assist farmers in deciding what type of produce to grow. In this report, it provides descriptive

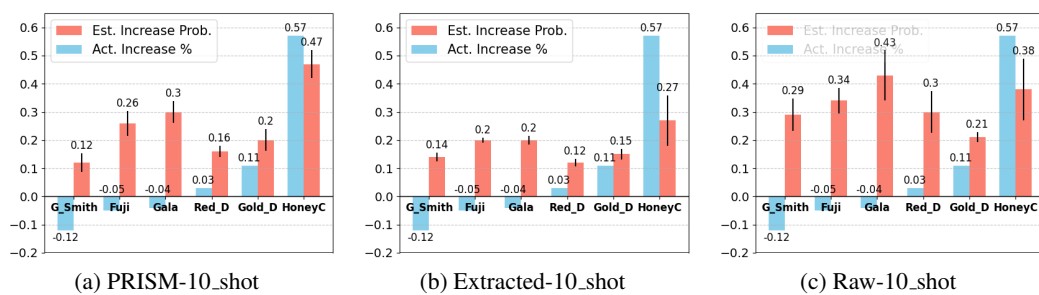

(a) PRISM-10_shot     (b) Extracted-10_shot     (c) Raw-10_shot

Figure 4: Prediction on whether the price of a type of apple will increase. Blue bars are actual price increase rates. Red bars are the estimated increase probability.

analysis regarding key factors that can influence apple prices. For each type of apple, we first use an LLM to generate summaries across seven aspects: production, demand, storage, imports and exports, policy, cost, and varietal competition. Each summary is then directly treated as a factor in PRISM. Notably, we use GPT-4.1 model for factor extraction and PRISM implementation, as its knowledge-cutoff is June 2024 and guarantees knowledge absent in 2025.

In Figure 4, for each type of apple, we report the actual price change rate ((price of 2025 - price of 2024) / price of 2024) (blue bars), and the estimated probability of "the price will increase in 2025" (red bars). We compare PRISM with direct LLM prompting, which conducts estimation on the extracted factors (Figure 4b) and on the raw report (Figure 4c). Note that the factor extraction from a long report is highly stochastic and it can greatly impact the estimation, we repeat the extraction and estimation process for 10 times and report the average estimation outcome.

From the result, we see PRISM can provide relatively more promising estimation result. In detail, if we focus on "Honeycrisp (HoneyC)", which has the largest increase in 2025, both PRISM and "Extracted" give it highest estimated increase probability, although "Extracted" has a huge variance in its predictions. For "Granny Smith (G_Smith)", which has the largest decrease, only PRISM can give it a lower expectation than other apple varieties. However, it is difficult to have more fine-grained comparison for PRISM and other strategies, e.g, to compare "Fuji vs Red Delicious (Red_D)". In Figure 8 and Figure 9 in Appendix C, we provide all the factor details for "Honeycrisp" and "Granny Smith", showing the prediction of PRISM is reasonable and easy to interpret.

**Predicting football matches**. In this study, we randomly choose 10 English football matches in 2025, and we leverage PRISM to estimate the probability of: "the home team will win". Here, we use GPT-5-mini, which is released before but relatively close to match dates. We collect pre-match reports (sourced from Footbal365) and extract key features from five aspects: squad quality, head-to-head records, recent form, player availability and fitness, and external conditions. Figure 5 shows the esti-

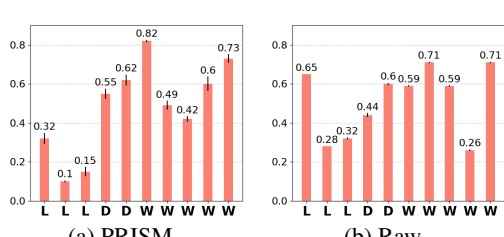

(a) PRISM     (b) Raw

Figure 5: Football match result prediction.

mated winning probability (**red bars**) and the match results (**x-axis**), where "L, D, W" denote "Lose, Draw and Win". Since the reports are relatively short, we only compare PRISM with LLM direct prompting from the raw texts. From the result, we can see PRISM can correctly predict the "Lose" cases by giving them a low prediction value, and it gives the two "Draw" matches winning probability around 0.5-0.6. For the "Win" games, it can tell 2 of out 5 winning matches by giving them scores over 0.7. Interestingly, we find PRISM tends to focus on accounting the factors themselves during prediction, while direct LLM prompting tends to rely on overall impressions, usually assuming the stronger teams are more likely to win (see the example in Figure 10 in Appendix C).

## 4.3 ABLATION STUDY

In this subsection, we further answer two important questions regarding PRISM: (1) Can it handle feature interaction (Hall, 1999)? (2) How is the computational efficiency of PRISM?

**Can PRISM handle feature interaction?** In ML probability estimation, feature interaction usually happens as the contribution of one factor to the outcome also highly depends on the condition

of another factor. Taking this into consideration is necessary for precise and reliable estimation. Figure 6 demonstrates that PRISM can indeed consider feature interactions when calculating the Shapley values and the final estimation. In detail, we focus on the task to predict loan default in Lending dataset under GPT-4.1-mini (see Section 4.1), where feature interactions can naturally occur between "Loan Amount" and "Annual Income". In Figure 6a, we compute the average Shapley value of Loan Amount across different instances, conditioned on loan amount (vertical axis) and annual income (horizontal axis). From the result, we can see that the individuals with annual income 120K+ receive a Shapley value of 0.18 for the factor "having a loan amount over 30K+". It is lower than the values assigned to people with income in 0–60K or 60K–120K. This indicates that, within PRISM, for individuals earning above 120K, having a loan amount over 30K+ **is not considered as risky as those with lower incomes**. Similarly, Figure 6b computes the average Shapley value of Annual Income. We can see PRISM believes that having a factor "Annual Income over 120K+" can greatly reduce the risk (-0.65), if their loan amount is over 30K+, and it only moderately reduces the risk (-0.26), if the loan amount is low, e.g., below 10K. It indeed shows a Shapely value in PRISM does not only rely on its factor of interest, but also other factors.

**Computational efficiency of PRISM.** In this part, we analyze the computational efficiency of PRISM. Table 3 summarizes two notions of complexity: *Query Complexity* counts the number of API requests that each method issues, and *LLM Evaluation Complexity* counts the number of instances that each method need to evaluate. In the table, we compare PRISM with 1-shot direct prompting and n-shot direct prompting. For PRISM, it needs to calculate Shapley values for m factors one by one. For each factor, it makes K samplings of background set for comparison. Therefore, it has a

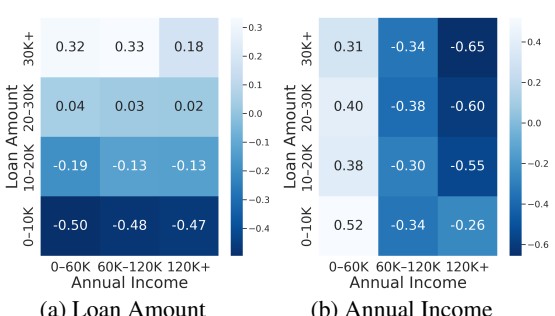

(a) Loan Amount    (b) Annual Income

Figure 6: Avg. Shapley under various conditions

query and evaluation complexity $\Theta(mK)$. Tabular-PRISM can input multiple S samplings into one query, so it has a query complexity $\Theta(m)$. In practice, we argue that Tabular-PRISM strategy can greatly reduce the time and token cost, as it saves API calls and evaluate multiple instances in one answer. For the actual time cost, in Stroke dataset (with $m = 10, K = 10$), Tabular-PRISM takes 92.7s for each instance on average under GPT-4.1-mini non-batched API calling. Apple price prediction using PRISM (GPT-4.1, $m = 7, K = 5$) takes around 330s on average, due to large amount of queries, long inputs (each factor is a paragraph) and larger model size. This efficiency can be acceptable if the evaluation size is not large.

| Metric | Query Complexity | LLM Evaluation Complexity |
|---|---|---|
| 1-shot | $\Theta(1)$ | $\Theta(1)$ |
| n-shot | $\Theta(n)$ | $\Theta(n)$ |
| Tabular-PRISM | $\Theta(m)$ | $\Theta(mK)$ |
| PRISM | $\Theta(mK)$ | $\Theta(mK)$ |

Table 3: Complexity of PRISM and baselines.

## 5 CONCLUSION AND LIMITATIONS

In this work, we propose Probability Reconstruction via Shapley Measures (RPSIM) for LLM-based probability estimation tasks. In our experiment, we empirically validate its predictive accuracy across multiple benchmark datasets, demonstrating the reliability of the proposed approach. Compared to direct LLM prompting, PRISM provides enhanced explainability and transparency, thereby enabling more trustworthy use of LLM predictions in high-stakes applications.

However, our work has a few limitations. First, it only focuses on the zero-shot setting. In practice, there could be historical records or references available to facilitate the prediction. In such few-shot prediction settings, interpretation can become more challenging, as the system need disentangle whether a prediction arises from its own knowledge or from the provided demonstrations. We leave this problem for future investigations. Second, our study focuses only on binary prediction problems. In practice, multi-class cases also arise, requiring a choice among several possible outcomes. While strategies like one-vs-all could extend our method, we did not test PRISM in such scenarios. Lastly, in Section 4.3, we examine the computational efficiency of PRISM and conclude that its cost can be relatively high, making it practical only when the evaluation size is not too large.

## 6 REPRODUCIBILITY STATEMENT

We release an anonymous repository with full source code and our processed datasets (if they are not publicly available) at https://anonymous.4open.science/r/prism-62B5/. In Appendix A, we provide the detailed algorithm sketches and theorem proof. Appendix B.1 and Appendix B.2 provide a complete description of our data pre-processing pipelines and the mentioned baselines. The experimental setup, including hyperparameters, model configurations are introduced in the main text. Appendix C contains additional examples, results and interpretations to aid further verification. Appendix D lists the exact prompts used for all language model components.

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

# A THEORY AND ALGORITHM

## A.1 DETAILED ALGORITHM

Guided by the Shapley additivity property (Property 1), PRISM estimates each factor's marginal effect via paired contrasts (realized vs. baseline value of that factor), averages these effects over randomly sampled contexts, and then *reconstructs* the model output, finally mapping to a calibrated probability.

**Setup.** Let $\mathcal{I} = \{1, \ldots, m\}$ index factors, $x = (x_1, \ldots, x_m)$ be the instance, $b = (b_1, \ldots, b_m)$ be designated baselines , $f$ be an evaluation oracle that returns either a *probability* in $[0, 1]$ or a *logit* in $\mathbb{R}$ when only a subset of factors is revealed, and $\Pi_i$ be a distribution over background sets $S \subseteq \mathcal{I} \setminus \{i\}$. Given $S$, we write $x_S$ for the partial specification that reveals $\{x_j : j \in S\}$.

---

**Algorithm 1** PRISM (Probability Reconstruction via Shapley Measures)

---

**Input:** Instance $x \in \mathcal{X}^m$; oracle $f$; sampling distributions $\{\Pi_i\}_{i=1}^m$; budget $K$.
**Output:** Probability $\hat{p}(x)$ and factor attributions $\{\hat{\phi}_i(x)\}$.
1: $\phi_0 \leftarrow f(x_\emptyset)$
2: **for** $i \in \mathcal{I}$ **do**
3:     $\Delta \leftarrow 0$
4:     **for** $k = 1$ to $K$ **do**
5:         Sample $S \sim \Pi_i$
6:         $\Delta \leftarrow \Delta + \big(f(x_{S \cup \{i\}}) - f(x_S)\big)$
7:     $\hat{\phi}_i(x) \leftarrow \Delta / K$
8: $\hat{z}(x) \leftarrow \phi_0 + \sum_i \hat{\phi}_i(x), \quad \hat{p}(x) \leftarrow \sigma(\hat{z}(x))$
9: **return** $\hat{p}(x)$ and $\{\hat{\phi}_i(x)\}$

---

**Algorithm 2** Tabular-PRISM (Batched realized vs. baseline contrasts)

---

**Input:** Instance $x \in \mathcal{X}^m$; oracle $f$; designated baselines $b$; sampling distributions $\{\Pi_i\}_{i=1}^m$; budget $K$.
**Output:** Probability $\hat{p}(x)$ and factor attributions $\{\hat{\phi}_i(x)\}$.
1: $\phi_0 \leftarrow f(x_\emptyset)$
2: **for** $i \in \mathcal{I}$ **do**
3:     For $k = 1, \ldots, K$, sample $S_i^{(k)} \sim \Pi_i$
4:     **Batch-Query:** evaluate all pairs

$$\left\{f(x_{S_i^{(k)} \cup \{i\}}|i := x_i), \ f(x_{S_i^{(k)} \cup \{i\}}|i := b_i)\right\}_{k=1}^K$$

5:     $\hat{\phi}_i(x) \leftarrow \frac{1}{K} \sum_{k=1}^K \left(f(x_{S_i^{(k)} \cup \{i\}}|i := x_i) - f(x_{S_i^{(k)} \cup \{i\}}|i := b_i)\right)$
6: $\hat{z}(x) \leftarrow \phi_0 + \sum_i \hat{\phi}_i(x), \quad \hat{p}(x) \leftarrow \sigma(\hat{z}(x))$
7: **return** $\hat{p}(x)$ and $\{\hat{\phi}_i(x)\}$

---

**Explanation for Alg. 1 .** (1) Query the oracle on the empty specification to obtain the intercept $\phi_0 = f(x_\emptyset)$. (2–3) Start looping over factors $i \in \mathcal{I}$ and initialize the accumulator $\Delta \leftarrow 0$ for factor $i$. (4–6) For $k = 1, \ldots, K$, draw a background set $S \sim \Pi_i$ and accumulate the presence/absence contrast $f(x_{S \cup \{i\}}) - f(x_S)$ into $\Delta$. (7) Average the $K$ contrasts to estimate the contribution of factor

i: $\hat{\phi}_i(x) \leftarrow \Delta/K$. (8) Reconstruct the score by additivity, $\hat{z}(x) = \phi_0 + \sum_i \hat{\phi}_i(x)$, and map through the logistic link to get the probability $\hat{p}(x) = \sigma(\hat{z}(x))$. (9) Return the probability and the per-factor attributions, i.e., $\hat{p}(x)$ and $\{\hat{\phi}_i(x)\}$.

**Explanation for Alg. 2 .** (1) Obtain the intercept by querying $f(x_\emptyset)$ so $\phi_0 = f(x_\emptyset)$. (2–3) For each factor i, sample K background contexts $S_i^{(k)} \sim \Pi_i$ to form the evaluation batches. (4) In each sampled context, evaluate a realized/baseline pair in batch: $f(x_{S_i^{(k)} \cup \{i\}} \mid i := x_i)$ and $f(x_{S_i^{(k)} \cup \{i\}} \mid i :=$ $b_i)$. (5) Average the K realized–baseline differences to obtain $\hat{\phi}_i(x) = \frac{1}{K} \sum_{k=1}^{K} \left(f(x_{S_i^{(k)} \cup \{i\}} \mid i := \right.$ $\left. x_i) - f(x_{S_i^{(k)} \cup \{i\}} \mid i := b_i)\right)$. (6) Reconstruct the score $\hat{z}(x) = \phi_0 + \sum_i \hat{\phi}_i(x)$ and apply the logistic link to produce $\hat{p}(x) = \sigma(\hat{z}(x))$. (7) Return $\hat{p}(x)$ together with the attributions $\{\hat{\phi}_i(x)\}$.

### A.2    THEOREM PROOF

**Proposition 1.** *Fix an instance* x *and a single reference sample* r. *Let* $\phi_i^{(r)}$ *be the Shapley value in Definition 2. Then, for models with deterministic scalar outputs, with* $\phi_0^{(r)} = v_r(\emptyset)$, *we still have:*

$$f(x) = v_r(\mathcal{I}) = \phi_0^{(r)} + \sum_{i=1}^{m} \phi_i^{(r)} \tag{7}$$

*Proof.* We use the permutation form of the Shapley value, which is equivalent to the subset form. Let $\pi$ be a permutation of $\mathcal{I}$, and let $\Pi(\mathcal{I})$ be the set of all m! permutations. For a permutation $\pi$ and an index i, let $\text{Pre}_i(\pi)$ denote the set of features that appear before i in $\pi$. The permutation form of the Shapley value of i for the game $v_r(S) = f([x_S, r_{\bar{S}}])$ can be written as

$$\phi_i^{(r)} = \frac{1}{m!} \sum_{\pi \in \Pi(\mathcal{I})} \left(v_r\big(\text{Pre}_i(\pi) \cup \{i\}\big) - v_r\big(\text{Pre}_i(\pi)\big)\right). \tag{8}$$

Taking equation (8) to the $\sum_{i=1}^{m} \phi_i^{(r)}$ part yields

$$\sum_{i=1}^{m} \phi_i^{(r)} = \frac{1}{m!} \sum_{\pi \in \Pi(\mathcal{I})} \sum_{i=1}^{m} \left(v_r\big(\text{Pre}_i(\pi) \cup \{i\}\big) - v_r\big(\text{Pre}_i(\pi)\big)\right). \tag{9}$$

For a fixed permutation $\pi = (\pi_1, \pi_2, \dots, \pi_m) \in \Pi(\mathcal{I})$, we have $\text{Pre}_i(\pi) \cup \{i\} = \text{Pre}_{i+1}(\pi)$ for $1 \le i \le m - 1$. Therefore

$$\sum_{i=1}^{m} \left(v_r\big(\text{Pre}_i(\pi) \cup \{i\}\big) - v_r\big(\text{Pre}_i(\pi)\big)\right) = v_r(\mathcal{I}) - v_r(\emptyset). \tag{10}$$

Substituting equation (10) into (9) gives

$$\sum_{i=1}^{m} \phi_i^{(r)} = \frac{1}{m!} \sum_{\pi \in \Pi(\mathcal{I})} \left(v_r(\mathcal{I}) - v_r(\emptyset)\right) = v_r(\mathcal{I}) - v_r(\emptyset). \tag{11}$$

Finally, given $\phi_0^{(r)} := v_r(\emptyset)$ and $v_r(\mathcal{I}) = f([x_\mathcal{I}, r_\emptyset]) = f(x)$, we have

$$f(x) = v_r(\mathcal{I}) = \phi_0^{(r)} + \sum_{i=1}^{m} \phi_i^{(r)}. \tag{12}$$

Proof complete. □

# B ADDITIONAL DETAILS FOR EXPERIMENTS

## B.1 DATASETS

**Stroke**: The stroke prediction dataset contains health, demographic, and lifestyle information for 5,110 patients, with the goal of predicting stroke occurrence. Each record includes variables such as age, hypertension, heart disease, marital status, work type, body mass index (BMI) and so on.

**Heart Disease**: The heart disease dataset integrates multiple clinical heart disease datasets and contains 918 patient records with 11 features related to demographics, clinical measurements, and lifestyle factors. The target variable indicates the presence or absence of heart disease.

**Adult Census**: The adult census contains 48,842 records from the 1994 U.S. Census, with 14 demographic and employment-related features such as age, education, occupation, work hours, and marital status. The target variable indicates whether an individual's annual income exceeds $50,000.

**Lending**: The lending dataset contains peer-to-peer loans issued from 2007–2018, with borrower and loan features such as income, debt-to-income ratio, FICO score, interest rate, loan amount, and purpose. The target variable indicates whether the loan will default.

Before applying above datasets for evaluation, we preprocess the datasets as follows:

(1) In order to make LLMs better understand the datasets, we rename the names of columns of datasets. For example, for **stroke** dataset, we rename "avg_glucose_level" as "average glucose level" and rename "bmi" as "Body Mass Index".

(2) For some columns, the values may be some abbreviation or with unclear meanings. For example, the values of attribute "gender" in dataset **stroke** are $\{0, 1\}$, so we convert "0" into "Female" and "1" into "Male". Besides, for attribute "Chest Pain Type" in dataset **heart disease**, the values are abbreviations such as "ATA", "NAP". We also convert them into their full names, e.g., "ATA" → "Atypical Angina" and "NAP" → "Non-Anginal Pain".

(3) We dropped some attributes of the datasets so that them LLMs can evaluation each data point with more efficiency.

The final data points examples and columns of each dataset are summarized in Figure (7).

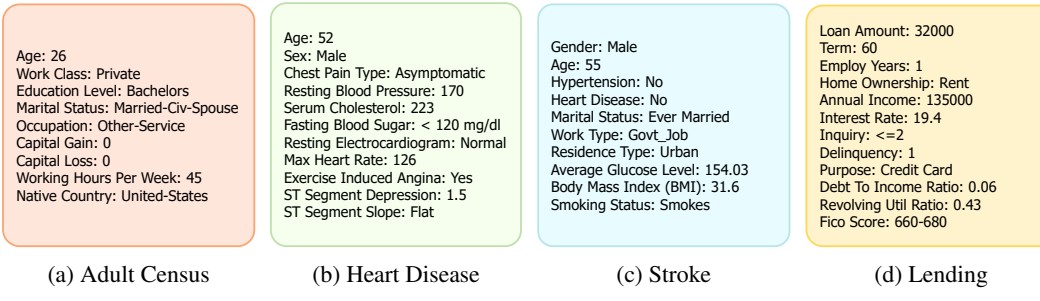

|        (a) Adult Census        |       (b) Heart Disease        |          (c) Stroke          |        (d) Lending         |

Figure 7: Data examples of the datasets.

## B.2 BASELINE METHODS

**1/5/10shot_level**: We perform multiple shots or multiple trials on directly asking LLM for probability as in prompt Figure (12). The LLM is instructed to select a linguistic probability description to represent the probability. Before evaluation, we map the descriptions into numerical values as follows: "very unlikely": 0.05, "unlikely": 0.2, "somewhat unlikely": 0.35, "neutral": 0.5, "somewhat likely": 0.65, "likely": 0.8, "very likely": 0.95. For 1shot_level, we only perform one trial; for 5shot_level and 10shot_level we perform 5 and 10 trials, and select the most common value as output.

**1/5/10shot_score**: We perform multiple shots similar to 1/5/10shot_level. The difference is that we instruct the LLM to output the numerical probability directly, as shown in the prompt Figure (13). We take the average probability as output.

**ICL-5+5, ICL-10+10**: We perform in-context learning on the 4 datasets with available training data. There are 5 positive 5 negative data in ICL-5+5 and 10 negative 10 positive data in ICL-10+10. The training data are excluding the evaluation data, with large mutual differences from the arbitrary candidate set. The order of the training data inside the prompt is shuffled randomly to minimize confusion. Detailed prompt example is shown as follows 14.

**Contrast**: We perform two queries in opposite directions. The first query apply the same prompt as in 1shot_level Figure (12). The second query is trying to ask the question in the opposite way, such as "How likely is this patient to NOT have a stroke?" instead of "How likely is this patient to have a stroke?". Two queries are normalized so that they sum up to 1, and the normalized positive answer will be the output.

**BIRD**: We follow BIRD's method with slight modification. First, since BIRD could not handle numerical datatype features, those features will be turned into bins before treated as input. In order to better utilize prior knowledge of LLM, binning will base on prior evidence that extract the most characteristic of the stage, therefore the interval of each bin may not have the same length. For example, regarding to the numerical feature "Resting Blood Pressure" in the Heart Disease dataset, the bins are: Normal '80-120', Pre-hypertension '120-130', Hypertension Stage 1 '130-140', Hypertension Stage 2 '140-180', Hypertensive crisis '180-200'. Second, instead of probability mapping in BIRD {"$f_j$ supports outcome i": 75%, "$f_j$ is neutral": 50%, "$f_j$ supports opposite outcome ¬i": 25%}, we are using a denser mapping which is the same as introduced in 1shot_level.

In general, the baseline BIRD is performed in 4 steps: (1) Initializing the prediction probability given each independent feature, by querying LLM using similar prompt as in 1shot_level Figure (12), only replace "Person information: ..." by "Given that [*factor*] = [*value*]". (2) Generating stochastic training data, by randomly choice a bin for each feature, the prompt is the same as in 1shot_level Figure (12). (3) Training the BIRD constrained optimization method. (4) Inferring to the evaluation dataset.

### B.3 REFERENCE INSTANCES SETTINGS

We instantiate a per-dataset *reference instance* $x_{ref}$ that serves as the baseline context for PRISM's contrastive evaluations. Reference values are chosen to be representative, i.e., close to the empirical mean for continuous variables and a prevalent category for discrete variables (the mode for multi-class features and the negative category for binary features). The concrete instances used in our experiments are listed in Table 4.

For each dataset's reference instance,, we use prompt 13, query the model five times, average the predicted probabilities, and convert the result to a base logit via logit(p) = $\log\left(\frac{p}{1-p}\right)$. The resulting base probabilities and logits are reported in Table 5.

## C ADDITIONAL RESULTS

### C.1 EXAMINATION OF INTERPRETATIONS

We examine factor-level attributions on the tabular datasets in Table 6, Table 7, and Table 8, and on the text scenarios in agriculture (Figure 8, Figure 9) and soccer (Figure 10).

**Adult (Table 6).** The attributions align with well-known socio-economic regularities for predicting income. Education exerts the largest and most consistent influence: *Masters* yields a strong positive contribution (Case 4: +1.31), while *HS-grad* is negative (Case 3: –0.45). Occupational roles are similarly informative: *Exec-managerial* is positive (Cases 1/4: +0.49/ + 0.99), whereas *Farming-fishing* is negative (Case 3: –0.84). *Capital gain* is highly predictive when present (Case 1: +1.04), and *age* contributes moderately with the expected direction (older age increasing odds in Cases 1/3). Marital status exhibits negative impacts for *Divorced* and *Never-married* (Cases 1/2), consistent with prior findings that marriage correlates with higher income. Some variables (e.g., *workclass*, *native country*) show near-zero effects in these cases, indicating either proximity to the anchor or low marginal power after conditioning on stronger factors.

**Heart (Table 7).** For heart disease risk, the factor impacts align with real-world clinical patterns *Exercise-induced angina* is strongly positive (Cases 1/2: +0.89/ + 0.87), and elevated *resting blood*

| Stroke | |
|---|---|
| Gender | Male |
| Age | 40.0 |
| Hypertension | No |
| Heart disease | No |
| Marital status | Never Married |
| Residence type | Rural |
| Average glucose level | 90.0 |
| Body Mass Index (BMI) | 24.0 |
| Work type | Private |
| Smoking status | never smoked |
| **Adult** | |
| Age | 40 |
| Workclass | Private |
| Education level | Some-college |
| Marital status | Married-civ-spouse |
| Occupation | Sales |
| Capital gain | 0 |
| Capital loss | 0 |
| Working hours per week | 40 |
| Native country | United-States |
| **Loan** | |
| Loan amount | 20000 |
| Term | 36 |
| Employ years | 3 |
| Home ownership | OWN |
| Annual income | 60000 |
| Interest rate | 14.0 |
| Purpose | car |
| Debt-to-income ratio | 0.35 |
| Revolving util ratio | 0.30 |
| FICO score | 680–710 |
| Inquiry | $\leq 2$ |
| Delinquency | 0 |
| **Heart Disease** | |
| Age | 53 |
| Resting Blood Pressure | 133 |
| Serum Cholesterol | 212 |
| Max Heart Rate | 137 |
| ST Segment Depression | 0.8 |
| Sex | Male |
| Chest Pain Type | Asymptomatic |
| Fasting Blood Sugar | $< 120$ mg/dl |
| Resting Electrocardiogram | Normal |
| Exercise Induced Angina | No |
| ST Segment Slope | Flat |

Table 4: Reference instance of each dataset.

| Dataset | p (mean over 5 runs) | logit |
|---|---|---|
| Stroke | 0.001 | -6.9068 |
| Adult | 0.354 | -0.6015 |
| Heart Disease | 0.410 | -0.3640 |
| Loan | 0.182 | -1.5029 |

Table 5: Base probabilities p (from prompt 13, averaged over five queries) and corresponding base logits per dataset.

*pressure* and *serum cholesterol* contribute positively (Case 1: +0.43/ + 0.45; Case 2: +0.38/ + 0.29). Conversely, higher *max heart rate* and *upsloping ST slope* decrease risk (Case 3: –0.45 and –0.48), aligning with cardiology practice that better exercise capacity and non-flat slopes are protective. *Sex=Female* is negative in Case 3 (–0.53), capturing lower risk in females. Note that the same feature

| Factor | Case 1 | | Case 2 | | Case 3 | | Case 4 | |
|---|---|---|---|---|---|---|---|---|
| | Value | Shapley | Value | Shapley | Value | Shapley | Value | Shapley |
| Age | 51 | 0.17 | 25 | -0.41 | 47 | 0.38 | 38 | 0.00 |
| Workclass | Private | 0.00 | Private | 0.00 | Self-emp-not-inc | -0.44 | Private | 0.00 |
| Education level | Bachelors | 0.62 | Bachelors | 0.49 | HS-grad | -0.45 | Masters | 1.31 |
| Marital status | Divorced | -0.47 | Never-married | -0.41 | Married-civ-spouse | 0.00 | Married-civ-spouse | 0.00 |
| Occupation | Exec-managerial | 0.49 | Sales | 0.00 | Farming-fishing | -0.84 | Exec-managerial | 0.99 |
| Capital gain | 10520 | 1.04 | 0 | 0.00 | 0 | 0.00 | 0 | 0.00 |
| Capital loss | 0 | 0.00 | 1876 | 0.00 | 0 | 0.00 | 0 | 0.00 |
| Working hours/wk | 40 | 0.00 | 40 | 0.00 | 60 | 0.54 | 60 | 0.53 |
| Native country | US | 0.00 | US | 0.00 | US | 0.00 | US | 0.00 |
| *Sum Shapley* | 1.86 | | -0.33 | | -0.81 | | 2.83 | |
| *Sum logit* | 1.26 | | -0.93 | | -1.41 | | 2.22 | |
| *Pred prob* | 0.778 | | 0.283 | | 0.197 | | 0.902 | |
| *True label* | Yes | | No | | No | | Yes | |

Table 6: Shapley values for four instances in Adult dataset. Reference instance: *age=40; workclass=Private; education=Some-college; marital status=Married-civ-spouse; occupation=Sales; capital gain=0; capital loss=0; working hours=40; native country=US*. A single base logit is shared across cases, $\phi_0 = \sigma^{-1}(0.354) = -0.6015$.

| Factor | Case 1 | | Case 2 | | Case 3 | | Case 4 | |
|---|---|---|---|---|---|---|---|---|
| | Value | Shapley | Value | Shapley | Value | Shapley | Value | Shapley |
| Age | 52 | 0.00 | 58 | 0.00 | 34 | -0.73 | 48 | -0.12 |
| Sex | Male | 0.00 | Male | 0.00 | Female | -0.53 | Male | 0.00 |
| Chest Pain Type | Asymptomatic | 0.00 | Non-Anginal Pain | 0.70 | Atypical Angina | 0.42 | Asymptomatic | 0.00 |
| Resting Blood Pressure | 170 | 0.43 | 150 | 0.38 | 118 | -0.49 | 132 | 0.00 |
| Serum Cholesterol | 223 | 0.45 | 219 | 0.29 | 210 | 0.00 | 272 | 0.37 |
| Fasting Blood Sugar | < 120 mg/dl | 0.00 | < 120 mg/dl | 0.00 | < 120 mg/dl | 0.00 | < 120 mg/dl | 0.00 |
| Resting ECG | Normal | 0.00 | ST-T abn. | 0.33 | Normal | 0.00 | ST-T abn. | 0.45 |
| Max Heart Rate | 126 | 0.19 | 118 | 0.00 | 192 | -0.45 | 139 | 0.00 |
| Exercise Induced Angina | Yes | 0.89 | Yes | 0.87 | No | 0.00 | No | 0.00 |
| ST Segment Depression | 1.5 | 0.44 | 0.0 | -0.59 | 0.7 | 0.00 | 0.2 | -0.44 |
| ST Segment Slope | Flat | 0.00 | Flat | 0.00 | Upsloping | -0.48 | Upsloping | -0.44 |
| *Sum Shapley* | 2.40 | | 1.99 | | -2.25 | | -0.63 | |
| *Sum logit* | 2.04 | | 1.62 | | -2.61 | | -0.54 | |
| *Pred prob* | 0.885 | | 0.835 | | 0.068 | | 0.367 | |
| *True label* | Yes | | Yes | | No | | No | |

Table 7: Shapley values for four instances in Heart dataset. Reference instance: *Age=53; Resting BP=133; Serum Chol.=212; Max HR=137; ST Depression=0.8; Sex=Male; Chest Pain=Asymptomatic; Fasting Blood Sugar=< 120 mg/dl; Resting ECG=Normal; Exercise Angina=No; ST Slope=Flat*. A single base logit is shared across cases, $\phi_0 = \sigma^{-1}(0.41) = -0.363$.

can change sign across cases (e.g., *ST depression*: –0.59 in Case 2 vs. near-zero/positive elsewhere), reflecting interactions and the conditional nature of $\phi_i(x)$ under different covariate settings.

**Loan (Table 8).** For default risk, the patterns are intuitive. High *interest rate* increases risk (Cases 3/4: +0.67/ + 0.71), while lower rates reduce it (Case 1: –0.50). Past *delinquency* is the single most influential positive factor when present (Case 3: +1.15). *Home ownership=RENT* and lower *annual income* tend to raise risk (Case 3: +0.44/+0.48), while higher *FICO* mitigates risk and lower *FICO* elevates it (Case 1: –0.41 vs. Case 3: +0.45). Debt burden is captured by *DTI* and *revolving utilization*: lower values reduce risk (Case 1: –0.08 and –0.51), whereas moderate-to-high levels are less favorable across other cases.

Across these tabular datasets, predicted probabilities exhibit a label-consistent ordering with clear separation—for example, the positives cluster at higher values while the negatives fall into intermediate and low ranges—yielding an easily interpretable ranking. Per-instance factor impacts make the drivers of each estimate explicit and support auditability: dominant contributors can be inspected, implausible probabilities traced to specific factors, and questionable cases flagged for review.

**Agriculture (Figure 8, Figure 9).** For Honeycrisp, *Production* contributes positively because a forecast output decline tightens supply, and this effect becomes stronger when *Storage* carryover is elevated and *Market demand* is soft. *Costs* add a small positive push as labor and inputs remain high, while *Varietal competition* subtracts; *Government policy* offers limited support. Together these effects yield a predicted probability of 0.51 for a price increase. For Granny Smith, the signs invert: an expected *Production* increase drives a negative contribution, which becomes larger under *Imports & exports* pressure and aggressive *Promotional activity*. The small positive *Government policy* term does not offset oversupply, and *Costs* together with *Varietal competition* further weigh on price. In this setting the predicted probability for a price increase is 0.09. Across the two varieties, these attributions accord with well-known agricultural price formation regularities in which supply shocks

| Factor | Case 1 (ID=1) | | Case 2 (ID=2) | | Case 3 (ID=29) | | Case 4 (ID=37) | |
|---|---|---|---|---|---|---|---|---|
| | Value | Shapley | Value | Shapley | Value | Shapley | Value | Shapley |
| Loan amount | 30000 | 0.45 | 1500 | 0.00 | 10000 | -0.38 | 30000 | 0.48 |
| Term | 60 | 0.45 | 36 | 0.00 | 36 | 0.00 | 60 | 0.51 |
| Employ years | 4 | -0.43 | < 1 | 0.45 | 6 | -0.43 | 10+ | -0.55 |
| Home ownership | MORTGAGE | 0.00 | MORTGAGE | 0.00 | RENT | 0.44 | MORTGAGE | 0.42 |
| Annual income | 65000 | -0.27 | 55000 | 0.00 | 48000 | 0.48 | 42000 | 0.46 |
| Interest rate | 12.0 | -0.50 | 10.4 | -0.43 | 20.0 | 0.67 | 19.4 | 0.71 |
| Inquiry | $\leq 2$ | 0.00 | $\leq 2$ | 0.00 | $\leq 2$ | 0.00 | $\leq 2$ | 0.00 |
| Delinquency | 0 | 0.00 | 0 | 0.00 | 1 | 1.15 | 0 | 0.00 |
| Purpose | Debt cons. | 0.43 | Home improv. | 0.00 | Debt cons. | 0.44 | Debt cons. | 0.48 |
| Debt-to-income ratio | 0.25 | -0.08 | 0.13 | -0.55 | 0.29 | 0.00 | 0.17 | 0.00 |
| Revolving util ratio | 0.21 | -0.51 | 0.29 | 0.00 | 0.08 | -0.47 | 0.44 | 0.00 |
| FICO score | 710–740 | -0.41 | 710–740 | -0.77 | 660–680 | 0.45 | 660–680 | 0.42 |
| *Sum Shapley* | -0.87 | | -1.30 | | 2.35 | | 2.93 | |
| *Sum logit* | -2.37 | | -2.81 | | 0.85 | | 1.43 | |
| *Pred prob* | 0.086 | | 0.057 | | 0.701 | | 0.806 | |
| *True label* | No | | No | | Yes | | Yes | |

Table 8: Shapley values for four instances in Loan dataset. Reference instance: *Loan amount=20000; Term=36; Employ years=3; Home ownership=OWN; Annual income=60000; Interest rate=14.0; Purpose=car; Debt-to-income ratio=0.35; Revolving util ratio=0.30; FICO score=680–710; Inquiry=$\leq$ 2; Delinquency=0.* A single base logit is shared across cases, $\phi_0 = \sigma^{-1}(0.182) = -1.504$.

and carryover inventory dominate short-run price movements, moderated by demand conditions, policy, and competing varieties.

**Soccer (Figure 10).** PRISM decomposes the match into conditional cues. *Home advantage* and *Squad quality* contribute positively (the latter is the largest, reflecting City's deeper roster), while *Head-to-head* contributes negatively (City have been troubled by Spurs) and *Player availability and fitness* is strongly negative given absences in midfield and doubts in key positions; *Recent form* is mildly positive and *External conditions* are neutral. These effects interact: the benefit of home advantage weakens when midfield anchors are missing, and the head-to-head penalty matters more when overall form is mixed. Balancing these factors yields a PRISM predicted probability of 0.44 for a City win at home (vs. 0.65 from direct LLM scoring), and the realized outcome—City lost—aligns more closely with PRISM's assessment.

Across agriculture and soccer, PRISM's factor impacts are conditional rather than global: identical features switch sign or magnitude as the surrounding evidence changes. This conditionality captures interaction structure among drivers, clarifies why conclusions can differ across otherwise similar factor sets, and yields interpretable, context-aware attributions aligned with domain regularities.

## C.2 CALIBRATION ANALYSIS

**Setup.** We assess calibration by plotting the weighted reliability curve. Given predictions $\{\hat{p}_i\}$ and labels $\{y_i \in \{0, 1\}\}$, we bin by *equal-count* quantiles of $\hat{p}$ and compute in-bin weighted means

$$\hat{p}_m = \frac{1}{W_m} \sum_{i \in B_m} w_i \hat{p}_i, \quad \hat{y}_m = \frac{1}{W_m} \sum_{i \in B_m} w_i y_i, \quad W_m = \sum_{i \in B_m} w_i. \tag{13}$$

Weights correct the gap between the *evaluation* split and the *deployment/population* class mix:

$$w_i = \begin{cases} \dfrac{\pi}{\hat{\pi}}, & y_i = 1, \\ \dfrac{1 - \pi}{1 - \hat{\pi}}, & y_i = 0, \end{cases} \qquad ECE = \sum_{m=1}^{M} \frac{W_m}{\sum_j W_j} |\hat{p}_m - \hat{y}_m|. \tag{14}$$

Here, $\hat{\pi}$ is the positive rate *in our evaluation split*, which is class-balanced ($\hat{\pi} = 0.5$). $\pi$ is the *population* positive rate we aim to evaluate against. Since the true deployment prevalence is unknown, we use the *original dataset prevalence before balancing* as a proxy for $\pi$: *Stroke* 4.87%, *Adult Census* 24.88%, *Heart Disease* 55.28%. (If a practitioner knows their deployment prevalence, they can plug it in for $\pi$.) We visualize calibration by plotting $(\hat{p}_m, \hat{y}_m)$ against the identity line $y = x$.

**Why weighting matters.** Our test splits are 1:1 balanced, but real-world prevalence is typically skewed. Without weighting ($w_i \equiv 1$), bin positive fractions reflect the artificial 50% mix rather than

*2025 Production** (Shapley value = 1.76)
- **Estimated Output:** For the 2024/25 crop year, U.S. Honeycrisp production is forecast at **27.7 million bushels** (42-lb bushels), down from 33.6 million in 2023/24. **This is a significant decrease (~17%),** likely a correction after last year's record crop.
- **Share of Total:** Honeycrisp will account for about **9.8% of total U.S. apple production**, making it the fourth largest variety after Gala, Red Delicious, and Granny Smith.
- **Trend:** Over the last five years, Honeycrisp production has been on the rise, but the 2025 forecast suggests a stabilization or slight pullback after a period of rapid expansion.

*2025 Market Demand** (Shapley value = -0.55)
- **Retail Price Pressure:** In 2023/24, Honeycrisp retail prices fell sharply (down 16% year-over-year for conventional), reflecting oversupply and aggressive discounting.
- **Premium Erosion:** Despite still commanding a premium over other varieties, the price gap is narrowing as production increases and retailers struggle to move large volumes.
- **Consumer Demand:** While Honeycrisp remains popular, **there are signs that demand is not keeping pace with supply,** especially at higher price points. The report notes that "retailers are finding it difficult to move that much fruit" - **Promotional Activity:** Heavy promotions in 2023/24 may not be sustainable, and there is concern that volumes sold at discount may not be repeatable in 2025.

*2025 Storage** (Shapley value = -1.09)
- **High Carryover:** In June 2024, **Honeycrisp storage holdings were **87% above** the previous year**, indicating slow movement and potential for significant carryover into the 2025 marketing year.
- **Storage Risk:** Large inventories increase the risk of price pressure and quality issues as the season progresses, especially if demand does not accelerate.
- **Packout Improvements:** Industry-wide, better storage protocols have improved packouts, but for Honeycrisp, this may mean even more fruit competing for limited shelf space.

*2025 Imports and Exports** (Shapley value = 0.0)
- **Exports:** U.S. apple exports rebounded in 2023/24, but there is no specific mention of Honeycrisp as a major export variety. Most U.S. apple exports are to Mexico, Canada, and Asia, but Honeycrisp's high price and quality requirements may limit its export share.
- **Imports:** Imports of fresh apples are down, and the U.S. is a net exporter. However, imported Honeycrisp is not a significant factor in the U.S. market.
- **Trade Policy Impact:** The return of the Indian market (after tariff removal) is positive for U.S. apples, but Honeycrisp is not a major variety for export to India or other recovering markets.

*2025 Government Policy** (Shapley value = 0.18)
- **Section 32 Purchases:** In 2023/24, the USDA made large Section 32 purchases to help absorb excess supply, including $56 million for fresh apples. Another $20 million buy is expected in fall 2024, which may help with Honeycrisp oversupply if included.
- **Labor Policy:** The industry is pushing for H-2A program reform to address high labor costs and shortages, which directly affect Honeycrisp growers due to the variety's labor-intensive harvest.
- **No Direct Subsidy:** There is no indication of Honeycrisp-specific government support, **but general apple industry interventions (purchases, labor reform) will indirectly benefit Honeycrisp growers.**

*2025 Costs** (Shapley value = 0.26)
- **Labor:** Labor is the largest cost, now accounting for ~60% of total production costs, **and is rising due to H-2A wage rates (over $18/hr in top states).**
- **Input Inflation:** Costs for fertilizer, utilities, and nursery trees have **all risen sharply (fertilizer up 92% over the decade).**
- **Margin Squeeze:** With retail and farm-gate prices falling and costs rising, many Honeycrisp growers are at risk of negative returns, especially if prices remain below cost of production.
- **Efficiency Pressure:** Growers are being forced to increase efficiency, but Honeycrisp's high input and labor requirements make this challenging.

*2025 Varietal Competition** (Shapley value = -0.51)
- **Rising Competition:** **New premium varieties like Cosmic Crisp® (up 3391% in five years), Pink Lady/Cripps Pink, and Envy® are gaining share and shelf space.**
- **Gala and Red Delicious:** While Gala is declining, it remains the top variety by volume and is rebounding in price, offering a lower-cost alternative to Honeycrisp.
- **Premium Segment Crowding:** The premium apple segment is becoming crowded, and Honeycrisp's unique selling proposition is being challenged by newer, often better-storing, and sometimes cheaper varieties.
- **Organic Competition:** Organic apples, including Honeycrisp, are holding price better than conventional, but the organic premium is also under pressure.

Figure 8: The factors and interpretations of PRISM for predicting whether the price of Honeycrisp apple will increase in 2025. Based on the Shapley values, it is finally predicted to have a chance of 51% to have price increase. Among the factors, "Production" has a large positive indication of price increase, as the production is expected to decrease by 17%.

---

*2025 Production** **(Shapley value = -0.74)**
- **Estimated Production:** For the 2024/25 crop year (the most current forecast, which covers the 2025 marketing season), U.S. Granny Smith production is projected at **28,252,464 bushels** (42-lb bushels), accounting for about **10% of total U.S. apple production**.
- **Trend:** **This is a slight increase from the previous year (27,762,121 bushels in 2023/24) and a notable rise compared to the five-year average (24,957,121 bushels). The data suggests Granny Smith is a stable or slightly growing variety in the U.S. mix.**
- **Regional Production:** The vast majority of Granny Smith apples are produced in Washington (25.9 million bushels), with smaller contributions from California, New York, Pennsylvania, and other states.

---

*2025 Market Demand** **(Shapley value = -0.25)**
- **Retail Price:** The average retail price for conventional Granny Smith apples in 2023/24 was **$1.58/lb**, down 9% from the previous year ($1.73/lb). This price is above Red and Golden Delicious, but below Honeycrisp.
- **Demand Trends:** The report notes a **long-term trend of declining per capita apple consumption,** especially among younger consumers. However, Granny Smith remains a popular variety due to its unique tart flavor and versatility (fresh eating, baking, processing).
- **Promotional Activity:** Aggressive retail promotions in 2023/24 likely boosted sales volume, but such volumes may not be sustainable if promotions are rolled back in 2025.
- **Varietal Position:** Granny Smith is the third most-produced variety, indicating continued strong demand, but faces increasing competition from newer, premium varieties.

---

*2025 Storage** **(Shapley value = -0.66)**
- **Storage Trends:** The report highlights that storage data is critical for understanding market movement. For Granny Smith, as with other major varieties, storage holdings are tracked monthly.
- **2023/24 Storage:** There was a general trend of slower movement for some varieties (e.g., Honeycrisp), but Granny Smith's storage movement is not specifically called out as problematic.
- **2025 Outlook:** With production up and prices down, **there is a risk of higher-than-normal storage holdings if demand does not keep pace, potentially leading to more late-season price pressure.**

---

*2025 Imports and Exports** **(Shapley value = -0.36)**
- **Exports:** The U.S. exported 46.4 million bushels of fresh apples in 2023/24, with Granny Smith being a significant export variety due to its global popularity. Key export markets include Mexico, Canada, Taiwan, and Vietnam.
- **Imports:** Imports of fresh apples into the U.S. were down 15% in 2023/24, and are expected to remain low in 2025 due to high domestic supply. Imports of Granny Smith specifically are not broken out, but most imported apples are from Chile, New Zealand, and Canada.
- **Trade Balance:** The U.S. maintains a strong positive trade balance in apples, **but the strong dollar and competitive global pricing may limit export profitability for Granny Smith in 2025.**
- **Export Challenges:** The report notes that U.S. exporters may be forced to sell at low margins to maintain market share, especially in price-sensitive markets.

---

*2025 Government Policy** **(Shapley value = 0.53)**
- **Labor Policy:** The H-2A Temporary Agricultural Worker Program is a major concern. Labor costs are rising, and the program is described as "untenable, unaffordable, and unsustainable." No major reform is expected by 2025, so labor will remain a key challenge.
- **Section 32 Purchases:** In 2023/24, the USDA made large Section 32 purchases to support the market. Another buy is expected in fall 2024, which may help absorb excess supply in 2025 if needed.
- **Trade Policy:** The removal of Indian tariffs on U.S. apples has reopened that market, but competition from low-cost suppliers (Iran, Türkiye) is fierce. U.S. policy focus is on maintaining and expanding export markets.
- **Cost Support:** No direct cost support is mentioned for Granny Smith, **but general industry support (e.g., for storage, export promotion) benefits all major varieties.**

---

*2025 Costs** **(Shapley value = -0.37)**
- **Production Costs:** Labor is the largest cost, accounting for about 60% of total production costs, and is rising due to higher Adverse Effect Wage Rates (AEWR). For top apple states, AEWR averages $18.22/hr, with California and Washington even higher.
- **Input Costs:** Fertilizer, utilities, and nursery trees have all seen significant cost increases (fertilizer up 92% over the last decade).
- **Margin Pressure:** While costs are up, grower prices for apples (including Granny Smith) have fallen sharply. In 2023/24, Granny Smith tray pack prices fell 45% from August to July.
- **Profitability:** **Many growers are expected to operate at a loss if these trends continue into 2025, especially if retail prices remain low and input costs high.**

---

*2025 Varietal Competition** **(Shapley value = -0.42)**
- **Rising Varieties:** **Honeycrisp, Cosmic Crisp®, and Pink Lady/Cripps Pink are gaining market share, often at the expense of older varieties like Red Delicious and, to a lesser extent, Gala.**
- **Granny Smith's Position:** While still a top-three variety, Granny Smith faces pressure from these newer, higher-priced varieties, especially in the fresh market.
- **Utilization:** Granny Smith is primarily a fresh-market apple (77% fresh, 23% processing), but is less likely to command the premium of Honeycrisp or Cosmic Crisp®.
- **Organic Competition:** Organic apples command a significant price premium and are holding their value better than conventional. Granny Smith is available as organic, but the conventional segment faces more price pressure.

---

Figure 9: The factors and interpretations of PRISM for predicting whether the price of Granny Smith apple will increase in 2025. Based on the Shapley values, it is finally predicted to have a chance of 9% to have price increase. Among the factors, only the "Government policy" is positive. In the negative factors, the production of Granny Smith is expected to increase, so it lowers the expectation of price increase.

Direct LLM Prompting predicted probability of Man City wins: 0.65

> Key points:
> - Home advantage + overall squad quality favors Man City (Haaland in form; City 4-0 last game).
> - Missing Rodri and Mateo Kovacic weakens City's midfield; Gvardiol and Foden doubts introduce uncertainty.
> - Tottenham also have significant absences (Kulusevski, Maddison) though Kudus/Richarlison are threats; Palhinha could add steel if he starts.
> - Recent head-to-head is mixed (Spurs have troubled City recently), but on balance City at home are still favorites.
> @@Final probability: 0.65@@

PRISM predicted probability of Man City wins: 0.44

> Squad Quality. **(Shapley value = 0.47)**
> Both teams boast strong attacking talent — City have Haaland (and promising Reijnders) and deeper overall quality; Spurs have dangerous forwards in Kudus and Richarlison but their squad was less consistent last season.

> Head-to-head: **(Shapley value = -0.41)**
> Spurs have troubled City recently — City have won only 4 of the last 12 meetings, including a 4-0 Spurs win last November and a 2-1 League Cup defeat; City did win 1-0 in February.

> Recent form: **(Shapley value = 0.15)**
> Good starts for both: City 4-0 vs Wolves, Spurs 3-1 vs Burnley — but both will face a sterner test here and City's poor run last season (six defeats in seven) is a reminder of volatility.

> Player availability and fitness: **(Shapley value=-0.45)**
> City: Kovacic and Rodri out; Gvardiol and Foden doubts; Rayan Cherki pushing for a start. Spurs: Kulusevski, Maddison and Dragusin out; Bissouma and Udogie doubtful; Joao Palhinha pushing for a start..

> External conditions: **(Shapley value=0.0)**
> Early Saturday lunchtime kick-off (typically low-key) but this fixture promises entertainment; the fixture computer gave both sides relatively easier openers, so this is a step up and will attract pundit attention.

Figure 10: Direct LLM Prompting and PRISM for estimating the probability of "Man City will beat Tottenham at Man City's Home". In general, Man City is a stronger team as it has a better squad. Direct LLM prompting yields facts similar to the factors used by PRISM, but concludes that Man City are favorites. This suggests that the LLM may rely on impressions to assign stronger teams higher winning probability. For PRISM, it also considers the factor "Squad Qualify" which favors Man City, but other factors such as head-to-head records and player availability lead PRISM give a lower winning probability for Man City—0.44. The result of the match is Man City loses.

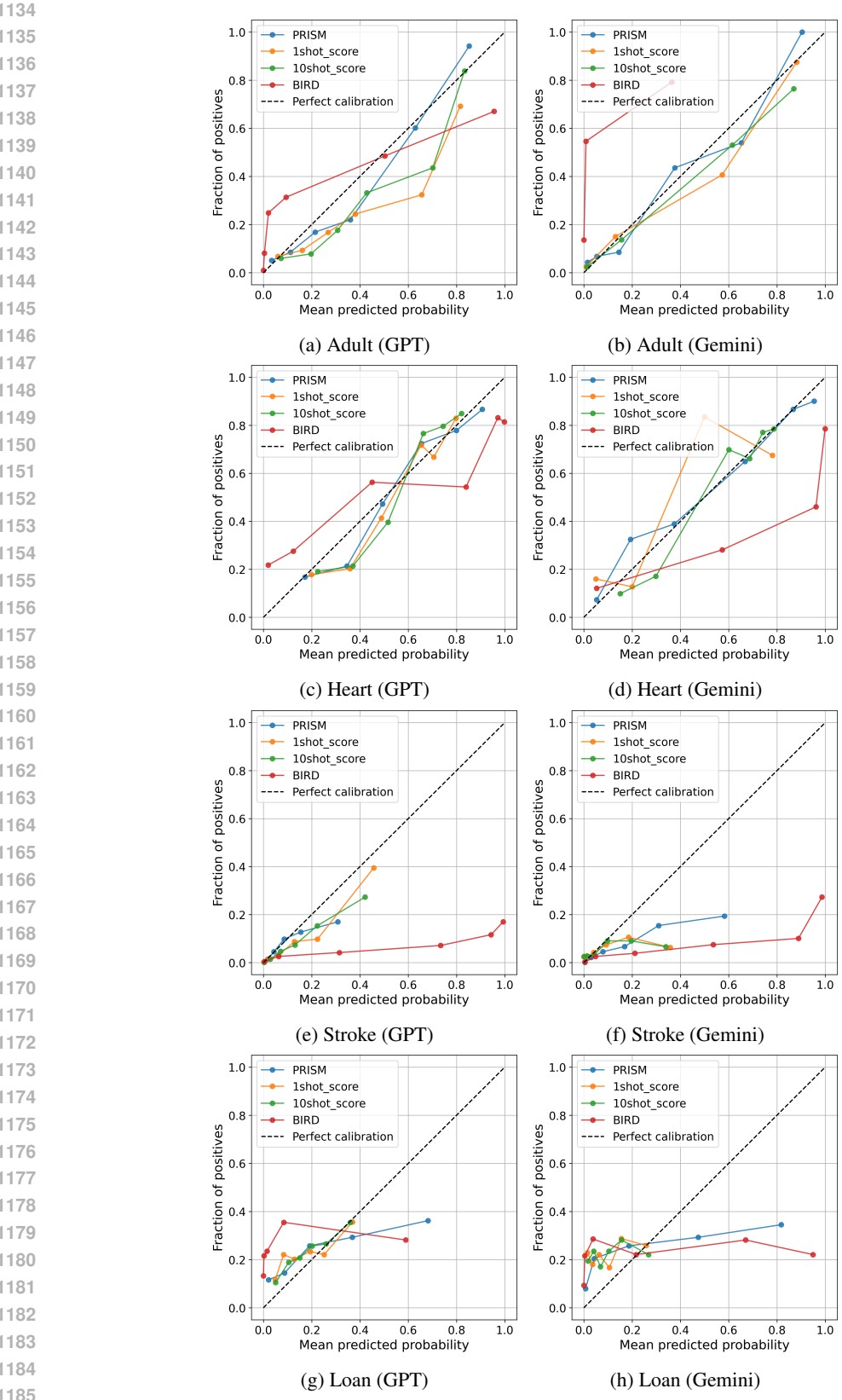

Figure 11: Calibration (reliability) curves comparing PRISM, 1shot_score, 10shot_score, and BIRD on four datasets under GPT-4.1-mini and Gemini-2.5-Pro.

the population, biasing the curve upward on imbalanced tasks (e.g., *Stroke*). Importance weights re-create the population mix *within each bin*, so the reliability curve answers the practical question: "given this score in deployment, what fraction will be positive?" Therefore, Reweighting restores the population class mix in each bin, yielding reliability curves that reflect real–world deployment rather than the artificial test mix.

**Results.** Across all four datasets in Fig. 11, *PRISM* maintains strong calibration, remaining close to the y=x line on *Adult* and *Heart*, and staying closest to the diagonal on *Stroke* and *Loan* despite small deviations, while *consistently yielding a monotonically increasing reliability curve*. This monotonicity ensures that higher predicted probabilities always correspond to higher empirical event rates (no local reversals), preventing rank inconsistencies and threshold instability that appear in the baselines when the fraction of positives decreases as the mean predicted probability increases, and it further shows that our method is better calibrated than the baselines.

# D  PROMPTS

```
You are an income prediction expert. Estimate the probability that the following person has an annual income
greater than $50,000.
This person lived in 1994; please base your judgment on the U.S. economic and social context of that year.

Person information:
[personal information in json]

How likely is this person to have income >$50,000?
First, provide a short explanation of your reasoning.
Then answer with one of: very unlikely, unlikely, somewhat unlikely, neutral, somewhat likely, likely, very likely.
```

(a) 1/5/10shot_level on Adult Census

```
You are a medical risk assessment expert. Estimate the probability that the following person will have heart
disease.

Person information:
[personal information in json]

How likely is this patient to have heart disease?
First, provide a short explanation of your reasoning.
Then answer with one of: very unlikely, unlikely, somewhat unlikely, neutral, somewhat likely, likely, very likely.
```

(b) 1/5/10shot_level on Heart Disease dataset

```
You are a medical risk assessment expert. Estimate the probability that the following person will have a stroke.

Person information:
[personal information in json]

How likely is this patient to have a stroke?
First, provide a short explanation of your reasoning.
Then answer with one of: very unlikely, unlikely, somewhat unlikely, neutral, somewhat likely, likely, very likely.
```

(c) 1/5/10shot_level on Stroke dataset

```
You are a loan-risk analyst. Estimate the probability that the following applicant will default on their loan.

Person information:
[personal information in json]

How likely is this applicant to be a defaulter?
First, provide a short explanation of your reasoning.
Then answer with one of: very unlikely, unlikely, somewhat unlikely, neutral, somewhat likely, likely, very likely.
```

(d) 1/5/10shot_level on Lending dataset

Figure 12: Prompts for baseline method 1shot_level, 5shot_level, 10shot_level. Where [personal information in json] is replaced by the actual data in json format.

# E  USE OF LARGE LANGUAGE MODELS

We only used large language models to assist with polishing the manuscript, and did not employ them for idea generation or any other substantive purpose.

```
You are an income prediction expert. Estimate the probability that the following person has an annual income
greater than $50,000.
This person lived in 1994; please base your judgment on the U.S. economic and social context of that year.

Person information:
[personal information in json]

How likely is this person to have income >$50,000?
First, provide a short explanation of your reasoning.
Then provide the final result strictly in the format:
##Final Result##: <a single number between 0 and 1>
```

(a) 1/5/10shot_score on Adult Census

```
You are a medical risk assessment expert. Estimate the probability that the following person will have heart
disease.

Person information:
[personal information in json]

How likely is this patient to have heart disease?
First, provide a short explanation of your reasoning.
Then provide the final result strictly in the format:
##Final Result##: <a single number between 0 and 1>
```

(b) 1/5/10shot_score on Heart Disease dataset

```
You are a medical risk assessment expert. Estimate the probability that the following person will have a stroke.

Person information:
[personal information in json]

How likely is this patient to have a stroke?
First, provide a short explanation of your reasoning.
Then provide the final result strictly in the format:
##Final Result##: <a single number between 0 and 1>
```

(c) 1/5/10shot_score on Stroke dataset

```
You are a loan-risk analyst. Estimate the probability that the following applicant will default on their loan.

Person information:
[personal information in json]

How likely is this applicant to be a defaulter?
First, provide a short explanation of your reasoning.
Then provide the final result strictly in the format:
##Final Result##: <a single number between 0 and 1>
```

(d) 1/5/10shot_score on Lending dataset

Figure 13: Prompts for baseline method 1shot_score, 5shot_score, 10shot_score. Where [personal information in json] is replaced by the actual data in json format, and the last line of each prompt is the format guideline.

```
You are an income prediction expert. Estimate whether a person in 1994 would have an annual income greater than
$50K, based on U.S. economic and social context of that year. Please use your own knowledge and the examples below
as references.
Examples:
- Features: [personal information in json]
  Label: [yes/no]
- Features: [personal information in json]
  Label: [yes/no]
…

Now predict the following case:
Features: [personal information in json]
Question: How likely is this person to have income $50,000?
First, provide a short explanation of your reasoning.
Then answer with one of: very unlikely, unlikely, somewhat unlikely, neutral, somewhat likely, likely, very likely.
```

(a) ICL on Adult Census

```
You are a medical risk assessment expert.
Estimate whether the following person will have a heart disease. Please use your own knowledge and the examples
below as references.
Examples:
- Features: [personal information in json]
  Label: [yes/no]
- Features: [personal information in json]
  Label: [yes/no]
…

Now predict the following case:
Features: [personal information in json]
Question: How likely is this patient to have a heart disease?
First, provide a short explanation of your reasoning.
Then answer with one of: very unlikely, unlikely, somewhat unlikely, neutral, somewhat likely, likely, very likely.
```

(b) ICL on Heart Disease dataset

```
You are a medical risk assessment expert. Estimate whether the following person will have a stroke. Please use your
own knowledge and the examples below as references.
Examples:
- Features: [personal information in json]
  Label: [yes/no]
- Features: [personal information in json]
  Label: [yes/no]
…

Now predict the following case:
Features: [personal information in json]
Question: How likely is this patient to have a stroke?
First, provide a short explanation of your reasoning.
Then answer with one of: very unlikely, unlikely, somewhat unlikely, neutral, somewhat likely, likely, very likely.
```

(c) ICL on Stroke dataset

```
You are a loan-risk analyst. Estimate the probability that the following applicant will default on their loan.
Please use your own knowledge and the examples below as references.
Examples:
- Features: [personal information in json]
  Label: [yes/no]
- Features: [personal information in json]
  Label: [yes/no]
…

Now predict the following case:
Features: [personal information in json]
Question: How likely is this applicant to be a defaulter?
First, provide a short explanation of your reasoning.
Then answer with one of: very unlikely, unlikely, somewhat unlikely, neutral, somewhat likely, likely, very likely.
```

(d) ICL on Lending dataset

Figure 14: Prompts for baseline method ICL-5+5, ICL-10+10. Where [personal information in json] is replaced by the actual data in json format. For ICL-5+5, we have 5 positive and 5 negative training data randomly ordered as Features and Values in the prompt, and similar for ICL-10+10 except we have 10 positive and 10 negative training data.

```
You are an income prediction expert. Estimate the probability that the following person has an annual incomeat most
$50,000.
This person lived in 1994; please base your judgment on the U.S. economic and social context of that year.

Person information:
[personal information in json]

How likely is this person to have income <=$50,000?
First, provide a short explanation of your reasoning.
Then answer with one of: very unlikely, unlikely, somewhat unlikely, neutral, somewhat likely, likely, verylikely.
```

(a) Contrast on Adult Census

```
You are a medical risk assessment expert. Estimate the probability that the following person will NOThave heart
disease.

Person information:
[personal information in json]

How likely is this patient to NOT have heart disease?
First, provide a short explanation of your reasoning.
Then answer with one of: very unlikely, unlikely, somewhat unlikely, neutral, somewhat likely, likely, verylikely.
```

(b) Contrast on Heart Disease dataset

```
You are a medical risk assessment expert. Estimate the probability that the following person will NOT have a stroke.

Person information:
[personal information in json]

How likely is this patient to NOT have a stroke?
First, provide a short explanation of your reasoning.
Then answer with one of: very unlikely, unlikely, somewhat unlikely, neutral, somewhat likely, likely, very likely.
```

(c) Contrast on Stroke dataset

```
You are a loan-risk analyst. Estimate the probability that the following applicant will NOT default on their loan.

Person information:
[personal information in json]

How likely is this applicant to be a non-defaulter?
First, provide a short explanation of your reasoning.
Then answer with one of: very unlikely, unlikely, somewhat unlikely, neutral, somewhat likely, likely, verylikely.
```

(d) Contrast on Lending dataset

Figure 15: Negative prompts for baseline method Contrast. Where [personal information in json] is replaced by the actual data in json format. The positive prompt is the same as in 1shot_label prompt12.

```
[A Markdown table with 10 pairs of records]

Your task is to evaluate the likelihood that each person has annual wage over 50K dollars in the United
States in 1994.

For each person:
- Conduct a brief analysis.
- Give a risk score from 1 to 10 (1 = very unlikely, 10 = very likely).

Provide the final result in the format ##Final Result##: @ID1: 1-10; @ID2: 1-10...
```

(a) Tabular-PRISM on Adult Census

```
[A Markdown table with 10 pairs of records]

Your task is to evaluate the likelihood that each person has heart disease.

For each person:
- Conduct a brief analysis.
- Give a risk score from 1 to 10 (1 = very unlikely, 10 = very likely).

Provide the final result in the format ##Final Result##: @ID1: 1-10; @ID2: 1-10...
```

(b) Tabular-PRISM on Heart Disease dataset

```
[A Markdown table with 10 pairs of records]

Your task is to evaluate the likelihood that each person has stroke.

For each person:
- Conduct a brief analysis.
- Give a risk score from 1 to 10 (1 = very unlikely, 10 = very likely).

Provide the final result in the format ##Final Result##: @ID1: 1-10; @ID2: 1-10...v
```

(c) Tabular-PRISM on Stroke dataset

```
[A Markdown table with 10 pairs of records]

Your task is to evaluate the likelihood that each loan application will default.

For each application:
- Conduct a brief analysis.
- Give a risk score from 1 to 10 (1 = very unlikely, 10 = very likely).

Provide the final result in the format ##Final Result##: @ID1: 1-10; @ID2: 1-10...
```

(d) Tabular-PRISM on Lending dataset

Figure 16: Prompt templates for Tabular–PRISM across four datasets. Each panel shows the standardized instruction used to score instances from a Markdown table with 10 pairs of records; each pair comprises a target case and its matched baseline case.

```
[2024 USApple Report]

Based on the report, estimate how likely the price of U.S. conventional [Apple variety's name] apple
will increase in the year 2024/2025. Briefly analyze and provide your answer in: @@Final Result: [0-
1]@@
```

(a) Raw

```
Based on the report, comment on the situation of U.S. conventional [Apple variety's name] apple,
covering the following aspects:
1. 2025 production. 2. 2025 market demand. 3. 2025 storage. 4. 2025 Imports and exports. 5. 2025
Government policy. 6. 2025 costs. 7. 2025 varietal Competition.
```

```
[Response from previous prompt]

Based on the report, estimate how likely the price of U.S. conventional [Apple variety's name] apple
will increase in the year 2024/2025. Briefly analyze and provide your answer in: @@Final Result: [0-
1]@@
```

(b) Extracted

```
Based on the report, comment on the situation of U.S. conventional [Apple variety's name] apple,
covering the following aspects:
1. 2025 production. 2. 2025 market demand. 3. 2025 storage. 4. 2025 Imports and exports. 5. 2025
Government policy. 6. 2025 costs. 7. 2025 varietal Competition.
```

```
We have information about [Apple variety's name] apple this year.
[Baseline Information]

Q1. Estimate the likelihood (1-10) that the price in 2024/2025 will be higher than this year.
- 1 = very unlikely, 10 = very likely.

Q2. New info: [Target factor from the previous prompt]
If the new info directly influences the estimation, give a new score. If no, provide the original
estimation.
Briefly explain and provide the final estimations in the format ##Final Result##: @Q1:[1-10] Q2:[1-10]@
```

(c) PRISM

Figure 17: Apple prompts of three experiments: *Raw* input, *Extracted* factors, and *PRISM*.

```
We have information of a football match between [Awayteam] at [Hometeam]'s home.

[Soccer Report]

Based on the report, estimate how likely the home team will win the match. Briefly analyze and provide
your answer in: @@Final probability: [0-1]@@.
```

(a) Raw

```
[Soccer Report]

Based on the report, briefly comment on the situation of this match, covering the following aspects:
1. Squad Quality. 2. Head-to-head records. 3. Recent form. 4. Player availability and fitness. 5.
External conditions. Provide the results following the format: 1) XXX \n\n2) XXX \n\n3) XXX \n\n4) XXX
\n\n5) XXX
```

```
We have information of a football match between [Awayteam] at [Hometeam]'s home.
[Baseline Information]

Q1. Estimate the likelihood (1-10) that the home team will win the match.
- 1 = very unlikely, 10 = very likely.

Q2. New info: [Target factor from the previous prompt]

If the new info directly influences the estimation, give a new score. If not, provide the original
estimation.
Briefly explain and provide the final estimations in the format ##Final Result##: @Q1:[1-10] Q2:[1-10]@
```

(b) PRISM

Figure 18: Soccer prompts of two experiments: *Raw* input and *PRISM*.

