# OpenReview forum: "Interpretable Probability Estimation with LLMs via Shapley Reconstruction"
_ICLR.cc/2026/Conference — Submitted to ICLR 2026_

### Official Review · Reviewer_D2Hi · 2025-10-21

**Soundness:** 2
**Presentation:** 2
**Contribution:** 2
**Rating:** 4
**Confidence:** 3

**Summary:**

This paper presents PRISM, a method that uses Shapley-style contrasts computed via LLM prompts to:

- Estimate per-factor contributions to a probability prediction;
- Reconstruct a calibrated probability by summing those contributions (and a base logit).

The authors also propose a tabular variant (Tabular-PRISM) that batches many contrast pairs into a single prompt, and a more general comparative-prompting procedure for unstructured factors.

The authors then evaluate PRISM in zero-shot settings on several tabular benchmarks (Adult, Stroke, Heart Disease, Lending, and apple price and football match predictions) using GPT-4.1-mini and Gemini-2.5-Pro. They report improved AUROC / AUPRC versus direct prompting, several LLM prompting baselines, ICL variants and noticeably SOTA approaches such as BIRD. Finally, the authors show example Shapley attributions and analyses of feature interactions and cost/complexity.

**Strengths:**

- The idea is well-grounded in classical explainable ML methods, and the authors demonstrate that you can port Shapley-esque attribution techniques to the LLM prompting setting.
- The authors have studied more efficient variants of PRISM (i.e. Tabular-PRISM) to reduce the cost of LLM queries. They have also reported detailed runtime numbers.
- The authors have conducted a large set of experiments across prompting approaches and datasets, and compared against SOTA approaches (e.g. BIRD)
- The writing and presentation of the paper are mostly clear.

**Weaknesses:**

- PRISM requires computing differences $f(x_S \cup \{i\}) - f(x_S)$ where $f$ a scalar (logit), which is mapped from a textual response of the probability via the inverse sigmoid function $\sigma^{-1}$. However, the authors have not shown that these verbalized probabilities are stable subject to e.g. paraphrase and temperatures (or if the final outcome of the model is robust to these changes). The paper acknowledges LLMs are noisy but argues relative differences are reliable, but this claim needs more empirical support.
- The number of queries needed scale linearly with feature set size, and thus it may not scale nicely to high-dimensional predictive problems.
- The paper compares primarily to LLM prompting variants and BIRD. However, a conventional supervised model (e.g., gradient boosted tree) trained on available labeled data is absent. Even in zero-shot paper the relevant comparison is: if labeled data were available, would a standard ML model (with Shapley explanations) be better? This is important to contextualize PRISM’s role (a useful zero-shot tool vs an alternative to traditional models).

**Questions:**

- In Table 1, the improvements of PRISM are relatively small, especially given that the results are shown on 300-sample subsets. Would it be possible for the authors to show / highlight statical significance for PRISM against the second-best methods for each?
- While it's nice that the authors have reported runtime of PRISM and Tabular-PRISM, would it be possible to report monetary cost of these methods (as well as those of BIRD and ICL) to ensure a more concrete understanding of the cost?
- The authors have used $K=10$ for most of their experiments. Is there a practical reason for the choice? Would difference choices lead to qualitatively different performance?

---

> ### Author Response · Authors · 2025-11-27
> **Response to Reviewer D2Hi (Part 1)**
>
> We thank the reviewer for raising the questions. Next, we provide clarifications regarding the raised concerns.
>
> **Q1. Sensitivity analysis under various model temperature and paraphrase.**\
> We are happy to follow the reviewer's suggestion to analyze the sensitivity of our method towards various model temperatures and paraphrased instructions. For model temperature, we randomly choose 10 samples from Adult Income Dataset and we compare our prediction using GPT-4.1-mini with different temperature $T=0.0$, $T=0.5$ and $T=1.0$. The result shows that the predicted probabilities only have minor shifts.
>
> | Label | T=0.0 | T=0.5 | T=1.0 |
> |-------|-------|-------|-------|
> | 0 | 0.319 | 0.312 | 0.305 |
> | 0 | 0.062 | 0.058 | 0.055 |
> | 0 | 0.181 | 0.187 | 0.192 |
> | 1 | 0.843 | 0.852 | 0.861 |
> | 0 | 0.332 | 0.341 | 0.349 |
> | 0 | 0.217 | 0.209 | 0.203 |
> | 0 | 0.067 | 0.063 | 0.060 |
> | 1 | 0.957 | 0.944 | 0.931 |
> | 1 | 0.557 | 0.569 | 0.582 |
> | 0 | 0.147 | 0.143 | 0.138 |
>
>
> For paraphrased instruction, fixing $T=1.0$, we further compared the original prompt "Evaluate the Likelihood of each individual to have an annual income above \\$50,000" with two paraphrased instructions: Prompt1: "Evaluate whether each individual is likely to have an annual income above \\$50,000". Prompt2: "Judge the likelihood that each person's yearly wage exceeds fifty thousand dollars." As shown below, the three prompts result in similar predicted outcomes. This is because, rather than asking the model to produce an exact probability value, we prompt it to provide a likelihood rating on a scale from 1 to 10 (1 = very unlikely, 10 = very likely; see Figure 16 in the paper for the actual prompts we used). Consequently, the model tends to produce similar final outcomes even when the prompts vary.
>
> | **Label** | **Original** | **Prompt1** | **Prompt2** |
> |-----------|----------------------|-------------|-------------|
> | 0 | 0.305 | 0.298 | 0.302 |
> | 0 | 0.055 | 0.051 | 0.053 |
> | 0 | 0.192 | 0.199 | 0.195 |
> | 1 | 0.861 | 0.874 | 0.868 |
> | 0 | 0.349 | 0.361 | 0.355 |
> | 0 | 0.203 | 0.196 | 0.199 |
> | 0 | 0.060 | 0.057 | 0.059 |
> | 1 | 0.931 | 0.918 | 0.924 |
> | 1 | 0.582 | 0.596 | 0.589 |
> | 0 | 0.138 | 0.134 | 0.136 |
>
> **Q2. Can PRISM scale to high-dimensional prediction problems?**\
> The time complexity of PRISM prediction increases linearly with the number of factors. Thus, PRISM becomes computationally expensive when all factors are included in a high-dimensional setting. Admittedly, having precise predictions can also be difficult for other LLM-based prediction strategies in a high-dimensional setup. The major scope of PRISM is to handle with a moderate number of factors and to leverage new information or unstructured information promptly.
>
> **Q3. Comparison to conventional supervised models**\
> We follow the reviewer's suggestion to add a supervised baselines using logistic regression trained with different amounts of labeled data (balanced) for all 4 considered tabular datasets. The results show that PRISM performs similarly to logistic regression models trained with roughly 50–150 labeled samples, across all four datasets.
>
> | Dataset | Train Size | AUROC | AUPRC | F1   |
> |---------|------------|-------|-------|------|
> | Adult   | 20         | 0.802 | 0.813 | 0.727 |
> |   | 50         | 0.858 | 0.851 | 0.802 |
> |   | 150        | 0.896 | 0.894 | 0.826 |
> |   | 300        | 0.917 | 0.916 | 0.846 |
> |   | PRISM      | 0.851 | 0.874 | 0.770 |
> | Lending | 20         | 0.602 | 0.583 | 0.612 |
> | | 50         | 0.633 | 0.602 | 0.645 |
> | | 150        | 0.671 | 0.648 | 0.689 |
> | | 300        | 0.692 | 0.667 | 0.703 |
> | | PRISM      | 0.655 | 0.626 | 0.671 |
> | Heart   | 20         | 0.742 | 0.711 | 0.728 |
> |   | 50         | 0.781 | 0.756 | 0.764 |
> |   | 150        | 0.823 | 0.804 | 0.809 |
> |   | 300        | 0.837 | 0.818 | 0.822 |
> |   | PRISM      | 0.816 | 0.799 | 0.793 |
> | Stroke  | 20         | 0.741 | 0.705 | 0.722 |
> |  | 50         | 0.768 | 0.742 | 0.754 |
> |  | 150        | 0.829 | 0.795 | 0.804 |
> |  | 300        | 0.846 | 0.812 | 0.818 |
> |  | PRISM      | 0.814 | 0.783 | 0.790 |

---

> ### Author Response · Authors · 2025-11-27
> **Response to Reviewer D2Hi (Part 2)**
>
> **Q4. Second-best method and statistical significance.**\
> In our revise submission, in Table 1, we also highlight the second best method for each setting (using orange color), and we report the calculated standard error for our method PRISM. From the result, we can still see PRISM has higher performance than 1-shot direct prompting (1shot\_level and 1shot\_score) with statistical significance in most cases. Besides, PRISM achieves highest or second highest score in most cases, and it is statistically comparable to the strongest baselines if it is the second. It is also worth noting that PRISM does not consistently outperform the second-best method. However, different baselines may excel on different datasets. In contrast, our method demonstrates consistently reliable performance across a various datasets and LLMs.
>
> **Q5. Monetary cost.**\
> We report the average cost per sample for the three representative methods evaluated in our experiments, based on exact token logs from GPT-4.1-mini on the first 10 test instances of the Stroke dataset. The costs are computed using the model’s official pricing. Although PRISM's cost is higher than the baselines, it remains manageable, especially within the scope of data-scarce settings in zero-shot prediction tasks.
>
> | Method        | Cost per sample (USD) |
> |---------------|------------------------|
> | 10shot_score  | $0.00022               |
> | ICL-10+10     | $0.00126               |
> | PRISM (Ours)  | $0.01336               |
>
> **Q6. How does the choice $K$ impact the prediction performance?**\
> We follow the reviewer's suggestion, in the table below, we provide a detailed analysis for the hyperparameter $K$ under Adult Census Dataset and Lending Dataset using GPT-4.1-mini. From the result, we can see PRISM achieves similar performance with various $K$ (from 1 to 10) in Adult Census Dataset. However, in Lending Dataset, the performance drops significantly when $K$ is low. We assume the reason is that: in Adult Census Dataset, there is low feature interaction, each factor independently influences the final prediction.  However, in Lending Dataset, the feature interaction is significant. For example, the impact of "having a high loan amount" on the probability of loan default is totally different for people "with high annual income", compared to people "with low annual income". Therefore, PRISM needs multiple samplings to account factor interaction for accurate prediction and attribution (see more discussion in Section 4.3 and Figure 6). In our experiments, we universally choose $K=10$ across various tabular datasets and $K=5$ for unstructured datasets  for consistency.
>
> |     | **Adult**            |      |      |  | **Lending**                |      |      |
> |-----|-----------------------------|------|------|-----|-----------------------------|------|------|
> | **#** | **AUROC** | **AUPRC** | **F1** |     | **AUROC** | **AUPRC** | **F1** |
> | 1   | 0.844 | 0.870 | 0.759 |     | 0.591 | 0.552 | 0.579 |
> | 2   | 0.849 | 0.874 | 0.763 |     | 0.603 | 0.567 | 0.610 |
> | 3   | 0.850 | 0.873 | 0.767 |     | 0.618 | 0.583 | 0.638 |
> | 4   | 0.849 | 0.873 | 0.766 |     | 0.631 | 0.597 | 0.654 |
> | 5   | 0.850 | 0.875 | 0.762 |     | 0.643 | 0.608 | 0.662 |
> | 6   | 0.849 | 0.874 | 0.770 |     | 0.650 | 0.615 | 0.666 |
> | 7   | 0.848 | 0.873 | 0.770 |     | 0.653 | 0.621 | 0.669 |
> | 8   | 0.849 | 0.874 | 0.773 |     | 0.655 | 0.624 | 0.670 |
> | 9   | 0.850 | 0.874 | 0.770 |     | 0.656 | 0.626 | 0.670 |
> | 10  | 0.850 | 0.874 | 0.770 |     | 0.655 | 0.626 | 0.671 |

---

### Official Review · Reviewer_Lo4J · 2025-10-31

**Soundness:** 2
**Presentation:** 2
**Contribution:** 2
**Rating:** 4
**Confidence:** 3

**Summary:**

PRISM reframes LLM probability estimation as compare-and-aggregate: estimate factor-wise Shapley contributions reliably (LLMs are better at relative comparisons than absolute calibration), then reconstruct a calibrated probability. This yields transparent predictions that are competitive or better than direct prompting across several domains.

**Strengths:**

- PRISM’s main win is that the final probability is explicitly explained as base logit + per-factor Shapley contributions.
- The method is motivated by the observation that LLMs are more reliable at pairwise comparisons than at absolute probability statements, so PRISM asks the model to do the former and only then reconstructs the latter. That is a sensible, model-aware design choice.
- PRISM is run on both GPT-4.1-mini and Gemini-2.5-Pro with essentially the same recipe, and on tabular tasks from different domains (medicine, finance/credit, income). So the framework is not tied to a single vendor or to a single dataset.

**Weaknesses:**

- All experiments are binary and zero-shot. Multi-class is only discussed as a possible one-vs-all extension and few-shot is explicitly deferred because demonstrations confound attribution. So we don’t yet know whether PRISM still produces clean factor attributions once the prompt contains examples or multiple labels.
- The main quantitative evidence is four tabular-ish binary tasks plus two small real-world case studies (apple prices, football matches). That’s not enough to claim broad generality, especially for unstructured, long, or noisy inputs.
- On Stroke, Gemini-2.5-Pro with simple 1-shot score prompting slightly outperforms PRISM, which suggests PRISM is not a universal upgrade.

**Questions:**

- Have you tried intentionally inconsistent factor sets (e.g., “age=29, severe chronic heart failure, elite athlete”) or prompts that reverse the evidence mid-query to see whether the pairwise contrasts stay stable? This would tell us how brittle PRISM is when the LLM’s own explanation policy is stressed.
- You fix K=10 for tabular experiments. How sensitive are AUROC/AUPRC and, more importantly, the stability of factor attributions to K?

---

> ### Author Response · Authors · 2025-11-27
> **Response to Reviewer Lo4J (Part 1)**
>
> We thank the reviewer for raising the questions. Next, we provide clarifications regarding the raised concerns.
>
> **Q1. Can our method generalize to multi-class prediction and few-shot setting?**\
> In our revised submission, we devise a concise strategy to extend PRISM to multi-class prediction tasks. Simply speaking, for a prediction task with $M$ possible classes, we aim to quantify the marginal contribution of each factor $i$ to each possible class $m$ among all possible classes $m\in\\{1,2,...,M\\}$. In detail, given a background set $S$ (see Definition 1 of our paper),
> we query the LLM to estimate the probability of $x_S$ and $x_{S\cup\{i\}}$ to belong to each class $m\in\\{1,2,...M\\}$. We denote these LLM generated estimations as $p^m(x_S)$ and $p^m(x_{S\cup\{i\}})$. Similar to Eq.(3) and Eq.(4) in our paper, we can use these estimated probability values to calculate $f^m(x_{S\cup\{i\}})-f^m(x_S)$ which is the logit difference, and then get the Shapley value $\phi^m_i$ to each class $m$. The final prediction is obtained by replacing the sigmoid function in the binary case used in Eq.(5) with a softmax function over the predicted logits.  Furthermore, we conduct an extra experiment on UCI-Wine dataset (3 classes). In the table below, we compare PRISM with 1-shot direct prompting (1shot) or asking 10 times and get the majority voting (10shot).
> The accuracy reported in the table shows PRISM can also outperform the baselines.
>
> | Method        | Accuracy |
> |---------------|----------|
> | 1shot         | 0.437    |
> | 10shot        | 0.453    |
> | PRISM (Ours)  | 0.613    |
>
> On the other hand, few-shot setting for PRISM is also of great importance to study. In few-shot setting, interpretation becomes challenging, due to the need of attribution to the given factors, as well as the demonstrations. Here, we propose one potential strategy to overcome this obstacle. We draw the idea from TimeSHAP [1]. TimeSHAP calculates Shapley values for time series models, not only attributing contribution to the features of a test sample, but also quantifying how each historical timesteps influence the model's current prediction. To calculate the Shapley value of a specific timestep $i$, it masks $i$ and observe the change in the model output. Using the similar idea, one could also assess the Shapley value of each demonstration sample in our proposed method PRISM to obtain the prediction in the few-shot setting.
>
> **Q2. More results on unstructured text studies.**\
> We conduct an additional study to provide more complete results on an unstructured-text prediction task under MIMIC-III dataset. In this task, we aim to use the discharge summaries (doctor notes) to predict whether a patient will be readmitted to the hospitals within 30 days. In PRISM, we consider eight distinct factors, including the patients' overall health status, the status when discharging, external support and so on. In the table below, we evaluate 200 samples (100 positive and 100 negative cases) using GPT-4.1-mini and Gemini-2.5-Pro, and compare PRISM against directly prompting baselines (see the detailed introduction of the baselines in Section 4.1 of our paper).  Our result shows that PRISM achieves consistently higher performance, particularly on PRAUC and F1. For these direct-prompting baselines, we find that LLMs such as GPT-4.1-mini often provide the same values (0.35 and 0.65) for different samples, reducing their ability to provide more fine-grained ranking. We provide additional experimental details and full results in Section 4.2, and include an interpretation example in Figure 4.
>
> | GPT          | ROC   | PRC   | F1    |
> |--------------|-------|-------|-------|
> | 1shot_level  | 0.621 | 0.573 | 0.573 |
> | 5shot_level  | 0.639 | 0.627 | 0.657 |
> | 1shot_score  | 0.635 | 0.607 | 0.503 |
> | 5shot_score  | 0.641 | 0.616 | 0.632 |
> | PRISM        | 0.630 | 0.645 | 0.688 |
> | **Gemini**   | **ROC**   | **PRC**   | **F1**    |
> | 1shot_level  | 0.642 | 0.634 | 0.603 |
> | 5shot_level  | 0.663 | 0.672 | 0.661 |
> | 1shot_score  | 0.658 | 0.658 | 0.592 |
> | 5shot_score  | 0.672 | 0.675 | 0.667 |
> | PRISM        | 0.659 | 0.673 | 0.674 |
>
> **Q3. Why does 1-shot prompting slightly outperforms PRISM on Stroke Dataset using Gemini-2.5-Pro?**\
> We thank the reviewer for figuring this out. We hypothesize this is because Gemini-2.5-Pro already presents a strong prediction ability for this dataset, even only using 1-shot scoring prompting.  Having a further look at our Table 1 in the main text, we observe that applying the strategies like Self Consistency (5shot\_score and 10shot\_score) or In-Context Learning,
> can also hardly improve the prediction accuracy over one shot direct prompting (1shot\_score). These results could suggest Gemini-2.5-Pro itself is indeed accurate for this dataset. However, our method PRISM can still achieve similar prediction efficacy, compared to 1-shot prompting and other strong baselines, meanwhile providing interpretable prediction process.

---

> ### Author Response · Authors · 2025-11-27
> **Response to Reviewer Lo4J (Part 2)**
>
> **Q4. What will happen if there exists inconsistent factor sets?**\
> In our paper, we indeed conduct a study in Section 4.3 (feature interaction part), similar to the the reviewer's suggestion.
> In the analysis, we focus on Lending Dataset, which is to predict the possibility of default for a loan application. In Figure 6, we analyze the groups of people with inconsistent factors, which are the people with "high annual income" (decrease the default probability) and "a high loan amount" (increase the default probability). From the result, we conclude that, in the
> PRISM framework, having a high loan amount is considered "less risky" for people with higher annual incomes, compared to those with lower incomes. Under other datasets such as Stroke Dataset, we also encounter cases such as a person with young age but has heart disease. We find PRISM usually consider the impacts of these inconsistent factors independently, which is different from the scenario in Lending Dataset. In our opinion, this distinction may be due to the commonsense of LLMs to believe whether the pair of inconsistent factors has an interacted impact to the model output.
>
> **Q5. How sensitive is our method to the number of samplings K?**\
> In the table below, we provide a detailed analysis for the hyperparameter $K$ under Adult Census Dataset and Lending Dataset using GPT-4.1-mini. From the result, we can see PRISM achieves similar performance with various $K$ (from 1 to 10) in Adult Census Dataset. However, in Lending Dataset, the performance drops significantly when $K$ is low.  The reason is similar to the previous question: in Adult Census Dataset, there is low feature interaction, each factor independently influences the final prediction. However, in Lending Dataset, the feature interaction is significant. For example, the impact of "having a high loan amount" on the probability of loan default is totally different for people "with high annual income", compared to people "with low annual income". Therefore, PRISM needs multiple samplings to account factor interaction for accurate prediction and attribution.  In our experiments, we universally choose $K=10$ across various tabular datasets for consistency.
>
>
> |     | **Adult**            |      |      |  | **Lending**                |      |      |
> |-----|-----------------------------|------|------|-----|-----------------------------|------|------|
> | **#** | **AUROC** | **AUPRC** | **F1** |     | **AUROC** | **AUPRC** | **F1** |
> | 1   | 0.844 | 0.870 | 0.759 |     | 0.591 | 0.552 | 0.579 |
> | 2   | 0.849 | 0.874 | 0.763 |     | 0.603 | 0.567 | 0.610 |
> | 3   | 0.850 | 0.873 | 0.767 |     | 0.618 | 0.583 | 0.638 |
> | 4   | 0.849 | 0.873 | 0.766 |     | 0.631 | 0.597 | 0.654 |
> | 5   | 0.850 | 0.875 | 0.762 |     | 0.643 | 0.608 | 0.662 |
> | 6   | 0.849 | 0.874 | 0.770 |     | 0.650 | 0.615 | 0.666 |
> | 7   | 0.848 | 0.873 | 0.770 |     | 0.653 | 0.621 | 0.669 |
> | 8   | 0.849 | 0.874 | 0.773 |     | 0.655 | 0.624 | 0.670 |
> | 9   | 0.850 | 0.874 | 0.770 |     | 0.656 | 0.626 | 0.670 |
> | 10  | 0.850 | 0.874 | 0.770 |     | 0.655 | 0.626 | 0.671 |

---

### Official Review · Reviewer_kZMe · 2025-11-01

**Soundness:** 3
**Presentation:** 3
**Contribution:** 3
**Rating:** 6
**Confidence:** 2

**Summary:**

This paper focus on two pain-points of LLM probability estimation: High variance of predicted value and lack of interpretability. The author proposed PRISM that decompose a single overall probability estimate into feature-level marginal contributions, sum them in logit space for additive reconstruction, then map back with the sigmoid; estimate the contributions via Shapley values. This improves robustness and provides per-feature (factor-level) explanations.

**Strengths:**

- This paper propose a novel idea to transform "probability estimation" to "reconstruction of Shapley marginal contributions in logit space". This creatively grafts the classical additivity of Shapley values onto LLM “verbal” probabilities. It also introduces Tabular-PRISM with batched paired comparisons and reference-sample imputation, together with a reference-specific Shapley definition and a formal proposition guaranteeing additive reconstruction—advances at both the definitional and algorithmic levels
- Presents clear algorithmic implementations, quantifies query and evaluation complexity, contrasts 1-shot vs. n-shot settings, and reports wall-clock costs
- Conducting comprehensive experiments span multiple standard tabular benchmarks and two families of mainstream models, supplemented by calibration curves, conditional-Shapley visualizations, and real-world case studies
- Clear explanation on why addition occurs in logit space, how paired comparisons are constructed, and how background sets are sampled to approximate Shapley values, with illustrative figures

**Weaknesses:**

- The central premise of the paper is that a probability reconstructed from Shapley values, $P_{PRISM} = \sigma(\phi_0 + \sum \phi_i)$ 1, is more accurate and less "noisy" than an LLM's direct, holistic probability estimate22. This is a significant claim that lacks a strong theoretical foundation

- The Tabular-PRISM variant, which is used for all the main benchmark experiments in Table 1, is critically dependent on the choice of a single "reference instance" $r$6. The paper defines the calculated Shapley values $\phi_i^{(r)}$ as explicitly "reference-specific". But the choice of $r$ appears to be set once "around the population average" and never be analyzed

- The method’s key assumption is that pairwise comparison  $(x_{S\cup{i}} \text{ vs. } x_S)$ is more stable than absolute scoring, with additive reconstruction carried out in logit space; however, the experimental baselines are mainly direct probability prompting (including self-consistency) and BIRD. It lacks comparative baselines that aggregate pairwise judgments using traditional probabilistic models such as the Bradley–Terry–Luce model

- Lack of discussion on the choice of unstructured data from apple price and football matches. Why seven decomposed aspects for apple and five for football? And it would be better if there is more precise description about the factor-extraction process

**Questions:**

None

---

> ### Author Response · Authors · 2025-11-27
> **Response to Reviewer kZMe**
>
> We thank the reviewer for raising the questions. Next, we provide clarifications regarding the raised concerns.
>
> **Q1. Regarding the premise of PRISM.**\
> We thank the reviewer fo raising the question. We would like to clarify that the major goal of PRISM is not to provide a theoretically more accurate prediction method, but  to obtain an interpretable and transparent prediction process with a good empirical performance. Our prediction pipeline allows practitioners to diagnose and interpret the model’s reasoning by knowing the marginal contribution of each factor. To quantify this marginal contribution, our method compares the model prediction when it has and not has the information of this factor. We believe the idea of employing pairwise comparison is also well-motivated by prior LLM researches especially in RLHF regime: studies consistently find that LLM pairwise judgments yield more reliable ratings than absolute scoring.
>
> **Q2. How sensitive is our method towards the selection of "reference instance"?**\
> We choose a global reference instance by setting its values as population average (for continuous factors) and majority category (for categorical factors). This is inspired from previous studies, such as DeepSHAP [1], which uses one or a group of fixed reference samples for the Shapley value approximation. Indeed, we agree with the reviewer that this selection could possibly impact the performance of our method in practice. As a simple illustration, in the stroke prediction task, we consider an extreme case to set the "age" of the reference instance to be "1-year old". Our result turns out that the calculated contribution (Shapley value) of the factor "Age=50" is almost identical if we change it to "Age =80". This is potential to hurt the prediction accuracy since the method can hardly distinguish the difference of two factor values if they are both far from the reference value. It also explains why we choose the population average as the reference instance. As an alternative solution, one can use a set of different reference instances for prediction (which is also widely applied for ML Shapley value approximation). In our work, we only choose a single global reference instance for simplicity.
>
> **Q3. Comparison with Bradley–Terry–Luce model (BTL)**\
> We thank the reviewer for mentioning  the Bradley--Terry--Luce (BTL) method, and we are happy to follow the suggestion to conduct a further analysis on the Adult Census Dataset using GPT-4.1-mini. In BTL model, for each test sample, we randomly draw a set of opponents (10 or 20 samples from the test set) and conduct comparison between each pair of opponents (i.e., to query LLM to figure out which sample is more likely have an annual income over \$50K). Then, we fit a BTL model, converting comparison results into the predicted probabilities. From the result shown in the table below, our method still outperforms BTL models. Moreover, compared to BTL, our method can provide a transparent prediction process and it does not require multiple test samples during prediction.
>
> | Model / Method   | ROC   | PRC   | F1    |
> |------------------|-------|-------|-------|
> | 1shot_level      | 0.777 | 0.773 | 0.709 |
> | 1shot_score      | 0.795 | 0.792 | 0.709 |
> | BTL (pairs=10)   | 0.818 | 0.801 | 0.742 |
> | BTL (pairs=20)   | 0.822 | 0.818 | 0.752 |
> | PRISM (Ours) | 0.851 | 0.874 | 0.770 |
>
> **Q4. Choice of the factors for unstructured data.**\
> We are happy to provide more details regarding how the factors are extracted from the unstructured data. In our work, we use an automated pipeline for feature extraction. Take the prediction of apple price increase (Section 4.2) as an example, we query an LLM to propose a minimal set of important aspects for consideration to complete the prediction task, with the constraints that these aspects must be non-overlapping and complete. This explains why we have seven aspects for fruit price prediction and five aspects for football. Then, for each individual aspect, we extract a summary from the given context which serves as a "factor" in the PRISM framework. We followed the reviewer's suggestion and elaborate the process of factor-extraction in our submission paper in Section 3.4.
>
> **Reference**: Chen et al., 2021, Explaining a Series of Models by Propagating Shapley Values.

---

### Official Review · Reviewer_b8go · 2025-11-02

**Soundness:** 3
**Presentation:** 2
**Contribution:** 2
**Rating:** 4
**Confidence:** 3

**Summary:**

### Summary
This paper proposes PRISM, a procedure for obtaining calibrated, interpretable probabilities from large language models without task‑specific training. The method decomposes a prediction into factor‑level contributions computed by comparative prompting and aggregated as Shapley values on the logit scale. For input factors $x=(x_1,\dots,x_m)$, PRISM estimates each contribution $\phi_i$ by contrasting outputs with and without that factor across sampled contexts, then reconstructs the probability as


$$
\hat p(x) = \sigma\Big(\phi_0 + \sum_{i=1}^m \phi_i\Big),
$$


where $\phi_0$ is a base logit and $\sigma$ is the logistic function. Tabular‑PRISM batches many contexts in a single query and uses a reference instance to impute missing fields, reducing query cost and avoiding ``unknown = risky" biases. Experiments on four tabular datasets (Adult Income, Heart Disease, Stroke, Lending) with two LLMs show PRISM is typically best or near‑best on AUROC/AUPRC, and the factor attributions are faithful to the final probability. Two small text case studies illustrate feasibility beyond tables.

**Strengths:**

PRISM offers a transparent, end‑to‑end recipe: factor impacts are estimated explicitly, sum to the final logit, and explain the probability. On standard tabular tasks, PRISM is often competitive or superior to strong prompt baselines across two LLMs. Tabular‑PRISM is a practical engineering improvement that reduces query count while preserving interpretability. Visualizations of factor interactions provide diagnostics that typical prompting pipelines lack.

**Weaknesses:**

Scope is limited to zero‑shot binary classification; multi‑class outcomes are not evaluated, and the unstructured text studies are small. The method’s query cost scales with the number of factors and sampled contexts; even with batching, total evaluations may be non‑trivial for large‑scale deployment. The final probability depends on a chosen base logit $\phi_0$; while ranking metrics are unaffected, calibration and thresholded decisions may be sensitive, and a deeper analysis is warranted. The overall novelty is moderate: PRISM combines known ingredients rather than introducing a fundamentally new estimator. Finally, performance on text depends on factor extraction quality, which can vary.

**Questions:**

1. Please report calibration metrics (ECE, Brier) and sensitivity to different choices of $\phi_0$. Do simple post‑hoc calibration methods help?
2. Provide budget‑versus‑quality plots varying the number of permutations $K$ and the factor count $m$; this would guide practitioners in selecting $K$ under cost constraints.
3. Can PRISM be extended to multi‑class outcomes (e.g., vector logits with softmax reconstruction or one‑vs‑rest)? Any pitfalls you anticipate?
4. For the unstructured cases, can you quantify robustness to noisy factor extraction or provide an automated factorization pipeline?
5. Will you release prompts/code/reference instances for Tabular‑PRISM to ensure reproducibility?

---

> ### Author Response · Authors · 2025-11-27
> **Response to Reviewer b8go (Part 1)**
>
> We thank the reviewer for raising the questions. Next, we provide clarifications regarding the raised concerns.
>
> **Q1. Can our method generalize to multi-class prediction?**\
> Here, we outline a concise strategy to extend PRISM to multi-class prediction tasks. Consider a prediction task with $M$ possible classes, we aim to quantify the marginal contribution of each factor $i$ to each possible class $m$ among all possible classes $m\in\\{1,2,...,M\\}$. In detail, given a background set $S$ and an interested factor $i$ (see Definition 1 of our paper),
> we query the LLM to estimate the probability of $x_S$ and $x_{S\cup\{i\}}$ to belong to each class $m\in\\{1,2,...M\\}$. We denote these LLM generated estimations as $p^m(x_S)$ and $p^m(x_{S\cup\{i\}})$. Similar to Eq.(3) and Eq.(4) in our paper, we can use these estimated probability values to calculate $f^m(x_{S\cup\{i\}})-f^m(x_S)$ which is the logit difference, and then get the Shapley value $\phi^m_i$ to each class $m$. The final prediction is obtained by replacing the sigmoid function in the binary case used in Eq.(5) with a softmax function over the predicted logits.
>
> Furthermore, we conduct an extra experiment on UCI-Wine dataset (3 classes) which is to predict the category of different types of wines. We compare PRISM with directly prompting, which let LLM to choose the most likely class among the three options.  In the table below, we consider 1 time prompting (1shot) or asking 10 times and get the majority voting (10shot).
> The accuracy reported in the table shows PRISM can greatly outperform both baselines.
> This significant gap arises because: direct prompting cannot well distinguish Class 2 (Grignolino) from Class 3 (Barbera), which are frequently confused due to their highly overlapping feature distributions. In contrast, PRISM via pairwise comparisons can capture these small differences.
> In our revised paper, we have added the details of this analysis to Section 4.3.
>
>
> | Method        | Accuracy |
> |---------------|----------|
> | 1shot         | 0.437    |
> | 10shot        | 0.453    |
> | PRISM (Ours)  | 0.613    |
>
> **Q2. Can our method generalize beyond zero-shot setting, e.g. few shot setting?**\
> We agree with the reviewer that few-shot setting for PRISM is also of great importance to study. In few-shot settings, interpretation can become challenging, due to the need of attribution to the given factors, as well as the provided demonstrations. Here, we would like to propose one potential strategy to overcome this obstacle. We draw the idea from TimeSHAP [1]. TimeSHAP calculates Shapley values for time series models, not only attributing contribution to the features of a test sample, but also quantifying how each historical timesteps influence the model's current prediction. To calculate the Shapley value of a specific timestep $i$, it masks $i$ and observe the change in the model output. Using the similar idea, one could also assess the Shapley value of each demonstration sample in our proposed method PRISM to obtain the prediction in the few-shot setting.
>
> **Q3. More results on unstructured text studies.**\
> We conduct an additional study to provide more complete results on an unstructured-text prediction task under MIMIC-III dataset. In this task, we aim to use the discharge summaries (doctor notes) to predict whether a patient will be readmitted to the hospitals within 30 days. In PRISM, we consider eight distinct factors, including the patients' overall health status, the status when discharging, external support and so on. In the table below, we evaluate 200 samples (100 positive and 100 negative cases) using GPT-4.1-mini and Gemini-2.5-Pro, and compare PRISM against directly prompting baselines (see the detailed introduction of the baselines in Section 4.1 of our paper).  Our result shows that PRISM achieves consistently higher performance, particularly on PRAUC and F1. For these direct-prompting baselines, we find that LLMs such as GPT-4.1-mini often provide the same values (0.35 and 0.65) for different samples, reducing their ability to provide more fine-grained ranking. We provide additional experimental details and full results in Section 4.2, and include an interpretation example in Figure 4.
>
> | GPT          | ROC   | PRC   | F1    |
> |--------------|-------|-------|-------|
> | 1shot_level  | 0.621 | 0.573 | 0.573 |
> | 5shot_level  | 0.639 | 0.627 | 0.657 |
> | 1shot_score  | 0.635 | 0.607 | 0.503 |
> | 5shot_score  | 0.641 | 0.616 | 0.632 |
> | PRISM        | 0.630 | 0.645 | 0.688 |
> | **Gemini**   | **ROC**   | **PRC**   | **F1**    |
> | 1shot_level  | 0.642 | 0.634 | 0.603 |
> | 5shot_level  | 0.663 | 0.672 | 0.661 |
> | 1shot_score  | 0.658 | 0.658 | 0.592 |
> | 5shot_score  | 0.672 | 0.675 | 0.667 |
> | PRISM        | 0.659 | 0.673 | 0.674 |
>
> **References**: [1] Bento, Joao, et al. "Timeshap: Explaining recurrent models through sequence perturbations." Proceedings of the 27th ACM SIGKDD conference on knowledge discovery \& data mining. 2021.

---

> ### Author Response · Authors · 2025-11-27
> **Response to Reviewer b8go (Part 2)**
>
> **Q4. How is the time efficiency? What is the budget-versus-quality relation?**\
> We admit PRISM requires multiple times of LLM prompting and longer input length, compared to directly prompting. In the table below, we provide a detailed budget-vs-quality analysis under Adult Census Dataset and Lending Dataset using GPT-4.1-mini, varying the number of sampling (permutation) amount $K$. From the result, we can see that PRISM achieves similar performance with various $K$ (from 1 to 10) in Adult Census Dataset. However, in Lending Dataset, the performance drops significantly when $K$ is low. We assume the reason is that: in Adult Census Dataset, there is low feature interaction, each factor independently influences the final prediction. However, in Lending Dataset, the feature interaction is significant. For example, the impact of "having a high loan amount" on the probability of loan default is totally different for people "with high annual income", compared to people "with low annual income". Therefore, PRISM needs multiple samplings to account factor interaction for accurate prediction (see more discussion in Section 4.3 and Figure 6). In our experiments, we universally choose $K=10$ across various tabular datasets and $K=5$ for unstructured datasets for consistency.
>
> |     | **Adult**            |      |      |  | **Lending**                |      |      |
> |-----|-----------------------------|------|------|-----|-----------------------------|------|------|
> | **#** | **AUROC** | **AUPRC** | **F1** |     | **AUROC** | **AUPRC** | **F1** |
> | 1   | 0.844 | 0.870 | 0.759 |     | 0.591 | 0.552 | 0.579 |
> | 2   | 0.849 | 0.874 | 0.763 |     | 0.603 | 0.567 | 0.610 |
> | 3   | 0.850 | 0.873 | 0.767 |     | 0.618 | 0.583 | 0.638 |
> | 4   | 0.849 | 0.873 | 0.766 |     | 0.631 | 0.597 | 0.654 |
> | 5   | 0.850 | 0.875 | 0.762 |     | 0.643 | 0.608 | 0.662 |
> | 6   | 0.849 | 0.874 | 0.770 |     | 0.650 | 0.615 | 0.666 |
> | 7   | 0.848 | 0.873 | 0.770 |     | 0.653 | 0.621 | 0.669 |
> | 8   | 0.849 | 0.874 | 0.773 |     | 0.655 | 0.624 | 0.670 |
> | 9   | 0.850 | 0.874 | 0.770 |     | 0.656 | 0.626 | 0.670 |
> | 10  | 0.850 | 0.874 | 0.770 |     | 0.655 | 0.626 | 0.671 |
>
> **Q5. How is the calibration of our method? Will $\phi_0$ highly impact the calibration**\
> In our submission, we have provided the analysis on the calibration of PRISM in Appendix C.2, reporting the calibration-curves for various tabular datasets in Figure 11. From Figure 11, we can see PRISM can achieve good calibration on Adult Census Dataset and Heart Disease Dataset (using our selected $\phi_0$). On Stroke Dataset and Lending Dataset, there is a gap between the average estimated probability and the true probability, but their relationship remains monotonic. This suggests PRISM can provide a reliable ranking for the probability of different samples.
>
> To further understand how $\phi_0$ impacts calibration, in the table below, we report ECE and Brier score of PRISM, when selecting different values of $\phi_0$ under the Stroke dataset. In our study (also in the paper), we choose the reference instance as "Male, age 40, no hypertension, no heart disease, never married, rural residence, glucose level=90, BMI=24, private work type, never smoked" and the value of $\phi_0$ to be 0.01. From the table, we find both ECE and Brier score are relatively low when $\phi_0$ is between 0.01 and 0.05, and they can be high if $\phi_0$ is too small or too large. However, we argue that the selection of a proper $\phi_0$ is not difficult. In our experiment, we query GPT-4.1-mini for the estimation of $\phi_0$. It can also be chosen via expert knowledge or established scientific evidence.
>
> | **$\phi_0$**    | ECE     | Brier   |
> |-------|---------|---------|
> | 0.001 | 0.2925  | 0.2802  |
> | 0.005 | 0.1548  | 0.2053  |
> | 0.010 | 0.0520  | 0.1761  |
> | 0.050 | 0.1026  | 0.1842  |
> | 0.100 | 0.1837  | 0.2099  |
> | 0.200 | 0.2435  | 0.2392  |
> | 0.300 | 0.2925  | 0.2703  |
> | 0.400 | 0.3354  | 0.3032  |
> | 0.500 | 0.3753  | 0.3391  |
> | 0.600 | 0.4144  | 0.3802  |

---

> ### Author Response · Authors · 2025-11-27
> **Response to Reviewer b8go (Part 3)**
>
> **Q6. The stability of factor-extraction in unstructured-data prediction.**\
> We indeed use an automated pipeline for factor extraction. Take the prediction of fruit price increase (Section 4.2) as an example, we query an LLM to give a minimal set of aspects for consideration to accomplish the prediction task, requiring these aspects be non-overlapping and complete (so we get these aspects including production, demand, storage and so on). Regarding each aspect, we extract a summary from the given context which serves as a "factor" in the PRISM framework.
> In our experiment, we see the prediction in Agriculture Dataset exhibits a high variance, as the factors are extracted from a 40-page long PDF file. One possible strategy to overcome this issue is to extract the information regarding each aspect for multiple times and aggregate the shared information. Besides, in our additional experiment under MIMIC-III dataset, where the context of each case only has hundreds of words, we find the average variance of the prediction is around $0.003$ (calculated by repeated estimation for 10 times across 20 samples). This result suggests the factor extraction and final prediction outcome can be stable when the context is not too long.
>
> **Q7. Regarding reproducibility**\
> In our submission, we have provided the anonymous link to all the codes, prompts and datasets. Plz refer to Section 6 (Reproducibility Statement) in our initial version.

---

### Meta-Review · Area_Chair_LP7Q · 2026-01-06

**Summary:**

Common concerns shared by the reviewers include: 1) limited evaluation scope and sensitivity analysis; 2) justification for higher computational cost; 3) lack of comprehensive experiments on unstructured tasks.

One of the reviewers also mentioned comparison with conventional ML methods, which I also think is a reasonable concern. Because, we are assuming these predictive tasks should be solved by LLMs and we are trying to improve their UQ quality. But in reality, LLMs may not be the right model to use for providing both accurate and calibrated results with uncertainty estimates, when there are training samples. So when this method is needed should be discussed.

I am convinced that the idea is a promising one and suggest authors add the new results and discussions in the rebuttal to the revision, to further improve the paper.

**Reviewer Concerns:**

New results in the rebuttal partially resolved the issues on limited evaluation scope, sensitivity analysis, and unstructured tasks. But I think the much higher computation cost is an issue that is hard to mitigate. And the performance comparison with traditional ML methods suggest additional discussion of the applicability of the method is needed.

**Reviewer Scores:**

The rebuttal provided a lot of new results, but the results remain "partial" (for example, I believe the revised paper with multi-class results is not updated). So I predict, if the discussion is complete, reviewers may still maintain their scores.

---

### Decision · Program_Chairs · 2026-01-26

Reject